   

# Sephin1 reduces TDP-43 cytoplasmic mislocalization and improves motor neuron survival in ALS models

Emmanuelle Abgueguen[1],†, Massimo Tortarolo[2],†, Laura Rouviere[3], Stefania Marcuzzo[5], Laura Camporeale[2], Alexandre Henriques[3], Laura Pasetto[2], Georgia R Culley[3], Valentina Bonetto[2], Anca Marian[4], Beatrice L Lejeune[1], Anne Visbecq[1], Giuseppe Lauria[6], Edor Kabashi[4], Noëlle Callizot[3], Caterina Bendotti[2], Pierre Y Miniou[1]

A pathological hallmark of ALS is the abnormal accumulation of misfolded proteins (e.g., TDP-43) and enlarged endoplasmic reticulum (ER), indicating ER stress. To resolve this stress, cells initiate the Unfolded Protein Response (UPR). However, unresolved stress leads to apoptosis. In ALS, UPR activation fails to resolve proteostasis impairment. UPR activation modulators, among them Sephin1, reduce protein aggregates and improve motor neuron survival in ALS models. We demonstrate that following glutamate intoxication, Sephin1 increases motor neuron survival by reducing mitochondria ROS production and extranuclear TDP-43. Sephin1 reduces abnormal splicing because of TDP-43 nuclear loss of function following oxidative stress. In SOD1[G93A] mice, Sephin1 treatment decreases TDP-43 in triton-insoluble fraction, improving motor neuron survival in spinal cord. Sephin1 improves motor neurons survival, motor function and survival of mutated TDP-43 transgenic zebrafish. Sephin1 improves motor neuron survival in ALS models by reducing TDP-43 cytoplasmic mislocalization and its toxicity. These findings open new therapeutic opportunities for Sephin1 in neurodegenerative pathologies with TDP-43 proteinopathy, including ALS.

## Introduction

Amyotrophic lateral sclerosis (ALS) is a rare and fatal adult-onset neurodegenerative disease characterized by the progressive loss of motor neurons in the cortex, brainstem, and spinal cord. Most cases are sporadic but ~10% of patients have an inherited familial form (Al-Khayri et al, 2024). The most common causative genes are hexanucleotide repeat expansion in *C9ORF72*, superoxide dismutase 1 (*SOD1*), TAR DNA-binding protein 43 (*TARDBP*, *TDP-43*) and fused in sarcoma (*FUS*) (Al-Khayri et al, 2024). Several cellular processes, which interplay with each other, are involved in ALS, such as glutamate excitotoxicity, oxidative stress, mitochondrial dysfunction, proteostasis impairment/protein aggregation, RNA binding protein dysfunction/RNA toxicity or neuromuscular junction (NMJ) failure (Al-Khayri et al, 2024).

The presence of enlarged endoplasmic reticulum (ER) in motor neurons and ER stress has been observed in pathological samples of ALS patients (Oyanagi et al, 2008; Sasaki, 2010; Lautenschlaeger et al, 2012). Upon detection of ER stress, the cell initiates the Unfolded Protein Response (UPR) pathway to reestablish proteostasis (Jeon et al, 2023). This cellular response is achieved through coordinated transcriptional and translational activities, by decreasing protein flux entering the ER and at the same time by increasing the transcription and translation of genes involved in the resolution of stress. This key cellular process is regulated by the phosphorylation of eukaryotic initiation factor 2α (eIF2α) which reduces the overall translation of proteins, whereas allowing the selective translation of key transcription factors that control the expression of foldases, ER chaperones, autophagy components and proteins involved in protein synthesis and redox control. Once cellular stress is resolved, the holophosphatase complex, composed of phosphatase PP1 catalytic subunit (PP1c) bound to the stress-inducible regulatory subunit PPP1R15A, dephosphorylates eIF2α to restore protein translation. However, when the stress is prolonged or too intense, the activation of UPR leads to apoptosis and cell death (Liu et al, 2024). In ALS, the presence of protein aggregates and ER stress markers indicate that the UPR activation is insufficient to resolve this proteostasis impairment (Jeon et al, 2023). Several studies have shown that the prevention of eIF2α dephosphorylation, either by genetic inactivation of PPP1R15A or by using phosphatase complex PPP1R15A/PP1c inhibitor such as guanabenz or Sephin1 (also named Icerguastat or IFB-088), reduces protein aggregates and improves motor neuron survival, motor performance, and survival in ALS animal models (Saxena et al, 2009; Tsaytler et al, 2011; Vaccaro et al, 2013; Jiang et al, 2014; Wang et al, 2014; Das et al, 2015).

[1]InFlectis BioScience, Nantes, France   [2]Research Center for ALS, Istituto di Ricerche Farmacologiche Mario Negri IRCCS, Milano, Italy   [3]Neuro-Sys SAS, Gardannes, France   [4]Translational Research for Neurological Disorders Lab, Institut IMAGINE, Paris, France   [5]Neurology 4 - Neuroimmunology and Neuromuscular Diseases Unit, Fondazione Istituto Neurologico Carlo Besta, Milan, Italy   [6]ALS Centre - 3rd Neurology Unit, Department of Clinical Neurosciences, Fondazione IRCCS Istituto Neurologico Carlo Besta, Milan, Italy

Correspondence: emmanuelleabgueguen@inflectisbioscience.com; pierreminiou@inflectisbioscience.com
†Emmanuelle Abgueguen and Massimo Tortarolo are co-authors

Among the aggregated proteins found in motor neurons of ALS patients, TDP-43 is present in about 97% of ALS patient brains (Bodansky et al, 2010; Ling et al, 2010; Tan et al, 2017), where it forms pathological inclusions of phosphorylated and ubiquitinated proteins in the cytoplasm of motor neurons (Neumann et al, 2006; Hasegawa et al, 2008; Igaz et al, 2008; Kametani et al, 2016; Buratti, 2018). Recent publications have shown that the levels of TDP-43 protein in extracellular vesicles (EVs) from cerebrospinal fluid (CSF) and plasma were significantly increased in ALS patients compared with healthy controls (Sproviero et al, 2018; Chatterjee et al, 2024; Kato et al, 2024). Whereas there has been some debate about whether TDP-43 pathology occurs in SOD1-associated ALS (Mackenzie et al, 2007; Tan et al, 2007), recent studies have identified cytoplasmic TDP-43 inclusions in motor neurons from SOD1-associated ALS patients (Okamoto et al, 2011; Jeon et al, 2019), and an increase in triton-insoluble fraction (TIF) of TDP-43 in the spinal cord at the end stage of the disease in SOD1$^{G93A}$ mice (Shan et al, 2009; Sumi et al, 2009; Marino et al, 2015; Jeon et al, 2019). TDP-43 is a RNA/DNA-binding protein (RBP) that regulates transcription, pre-mRNA splicing, and translation, and is localized predominantly in the nucleus in normal physiological conditions (de Boer et al, 2020). In response to cellular stress such as ER stress or oxidative stress, TDP-43 can form cytoplasmic stress granules (SGs), which safeguard the essential mRNAs from degradation and promote rapid recovery after stress removal (Loganathan et al, 2019). However, under prolonged stress, TDP-43 protein inside SGs transitions from a liquid-like droplet to gel-like inclusion, inhibiting its ability to dissociate, leading to its accumulation into the cytoplasm (Yan et al, 2025). Increasing evidence suggests that the decrease in nuclear localization and the increase in cytoplasmic localization of TDP-43 induce toxicity through both loss and gain of function mechanisms. Loss of nuclear TDP-43 function leads to abnormal RNA splicing and incorporation of erroneous cryptic exons which provokes the loss of crucial neuronal proteins such as Stathmin-2 or UNC13A (Mehta et al, 2023; Koike, 2024). Elevated cytotoxicity is observed in cells and animal models expressing cytoplasmic TDP-43, whereas preventing TDP-43 nuclear export partially reduces TDP-43 cytotoxicity (Kabashi et al, 2010; Diaper et al, 2013; Walker et al, 2015; Cascella et al, 2016), indicating that cytoplasmic TDP-43 could promote TDP-43 cytotoxicity. Recent studies have shown that cortical hyperexcitability and glutamate excitotoxicity, both observed in ALS, induce TDP-43 translocation into the cytoplasm (Boussicault et al, 2020; Weskamp et al, 2020; Dyer et al, 2021; Berthiaume et al, 2024). Therefore, regulating TDP-43 localization, nuclear function and reducing TDP-43 cytoplasmic accumulation could be beneficial in the treatment of ALS.

Recent publications have shown that guanabenz, an FDA approved drug for hypertension, and its close derivative devoid of $\alpha$2 adrenergic receptor agonist activity, Sephin1, are beneficial in ALS animal models (Vaccaro et al, 2013; Jiang et al, 2014; Wang et al, 2014; Das et al, 2015). Furthermore, a phase 2 clinical trial in ALS patients suggested that guanabenz can slow down disease progression in bulbar-onset ALS patients (Dalla Bella et al, 2021), and a phase 2 clinical trial has just been completed to evaluate Sephin1 in bulbar-onset ALS patients (NCT05508074). Both small molecules have been described as protecting cells against ER stress by modulating the UPR pathway and reducing proteostasis dysfunction through the

inhibition of the stress induced phosphatase complex, PPP1R15A/PP1c (Tsaytler et al, 2011; Das et al, 2015; Carrara et al, 2017); however, this mechanism of action is controversial (Crespillo-Casado et al, 2017, 2018). In NMDA-intoxicated neurons, both guanabenz and Sephin1 prevent NMDA-induced neuronal cell death and partially reduce cytosolic Ca$^{2+}$ increase following NMDA stimulation, independently of the activation of the translation arm of the UPR pathway (Ruiz et al, 2020). In SOD1$^{G93A}$ mice, in which ER stress has been shown, guanabenz and Sephin1 have both been shown to decrease SOD1 aggregates, increase motor neuron survival and improve motor function (Jiang et al, 2014; Wang et al, 2014; Das et al, 2015). However, their impact on SOD1$^{G93A}$ mice survival was not clear (Jiang et al, 2014; Wang et al, 2014; Vieira et al, 2015, 2024). In mutant TDP-43 zebrafish embryos, guanabenz was shown to improve motor function by reducing ER stress (Vaccaro et al, 2013). Despite the evidence that Sephin1 could improve proteostasis, motor neuron survival and motor function, no data currently demonstrates that Sephin1 could reduce TDP-43 cytoplasmic localization or TDP-43 toxicity in ALS models.

This study aims to investigate whether Sephin1 could reduce TDP-43 cytoplasmic localization or TDP-43 toxicity and therefore protect motor neurons and increase survival in ALS models. First, the impact of Sephin1 on TDP-43 toxicity, in particular its localization, was evaluated in motor neurons following glutamate excitotoxicity. Indeed, glutamate intoxication in primary motor neurons from wild-type (WT) rat or from SOD1$^{G93A}$ transgenic rat leads to TDP-43 cytoplasmic mislocalization (Boussicault et al, 2020; Berthiaume et al, 2024). Second, the impact of Sephin1 on abnormal splicing due to loss of nuclear TDP-43 function was evaluated in human cell line, SH-SY5Y following oxidative stress. Recently, it was demonstrated that abnormal splicing due to the loss of TDP-43 nuclear function occurs during the arsenite-induced stress recovery (Huang et al, 2024). Third, Sephin1 was evaluated in two in vivo ALS animal model. The impact of Sephin1 on motor neuron survival and TDP-43 aggregates was evaluated in the SOD1$^{G93A}$ mouse model, one of the commonly used animal models for studying ALS and used to inform many clinical trials. SOD1$^{G93A}$ mice exhibit signs of ER stress, an abnormal glutamate release, cytoplasmic TDP-43 mislocalization and an increase of TDP-43 in TIF of the spinal cord (Milanese et al, 2011; Das et al, 2015; Marino et al, 2015; Bonifacino et al, 2016; Jeon et al, 2019). The impact of Sephin1 on TDP-43 toxicity was also evaluated in the TDP-43 transgenic zebrafish model in which motor neuron loss, motor dysfunction, and reduced survival have been shown (Lissouba et al, 2018).

We show that in primary WT and SOD1$^{G93A}$ motor neurons, Sephin1 reduces TDP-43 cytoplasmic localization and increases motor neuron survival following glutamate excitotoxicity in an eIF2$\alpha$ phosphorylation-independent manner. During the recovery phase after oxidative stress, Sephin1 reduces the expression of abnormally spliced mRNAs. In the two ALS animal models, SOD1$^{G93A}$ mice and mutant TDP-43 transgenic zebrafish embryos, Sephin1 improves motor neuron survival. In the spinal cord of SOD1$^{G93A}$ mice, Sephin1 reduces TDP-43 level in TIF of the spinal cord. In mutant TDP-43 transgenic zebrafish embryos, Sephin1 improves motor function and the survival of these animals indicating strong preclinical evidence for Sephin1 as a therapeutic strategy in ALS patients.

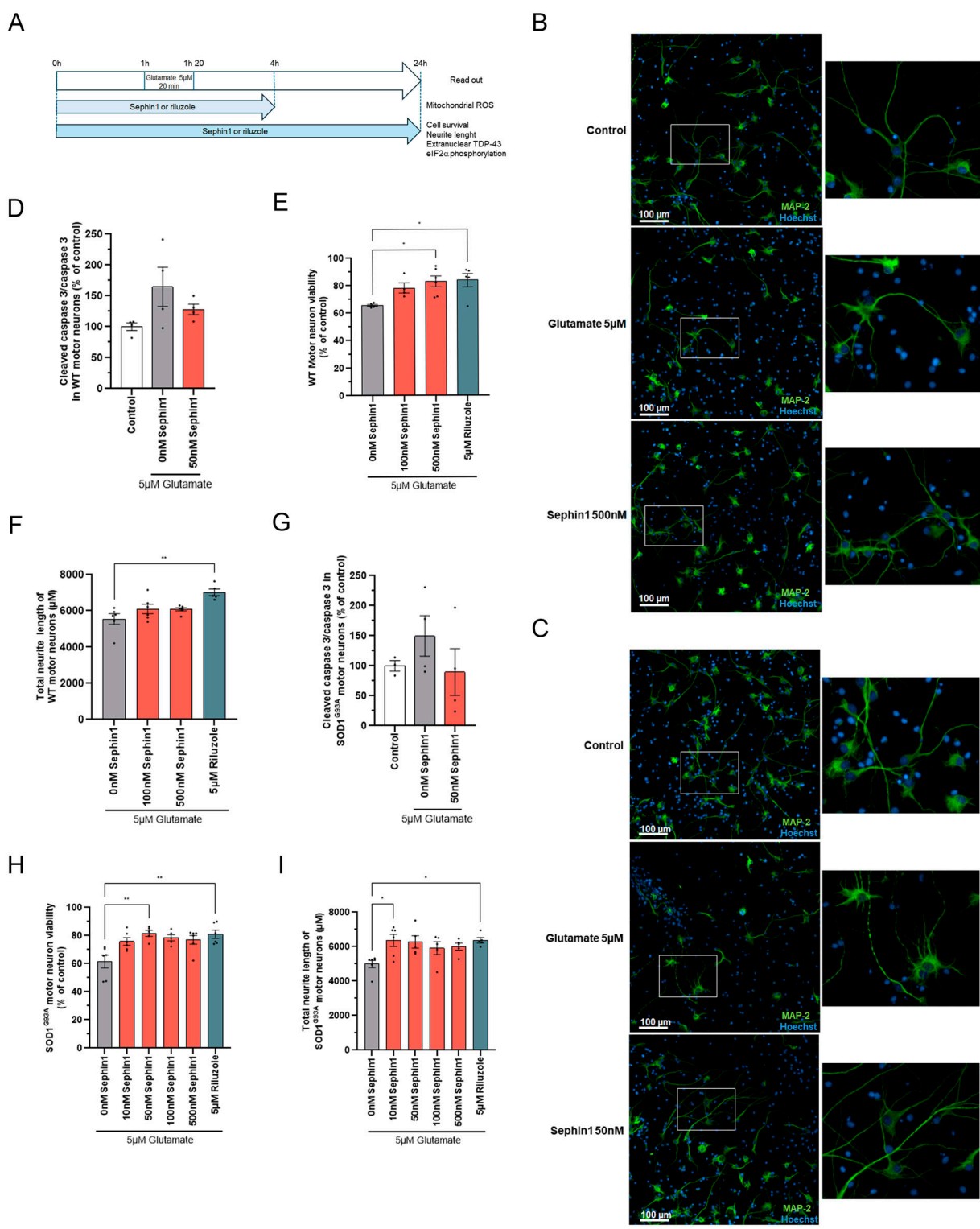

**Figure 1. Sephin1 protects primary WT and SOD1^G93A rat motor neurons against glutamate injury.**

**(A)** Schematic of the experiment on primary WT and SOD1^G93A rat motor neurons. **(B)** Representative pictures of primary WT rat motor neurons treated with glutamate 5 $\mu$M for 20 min in presence of DMSO (vehicle) or Sephin1 500 nM 24 h after glutamate intoxication. Motor neurons are labeled with the neuronal marker, MAP-2. Right panel: enlargement of motor neurons **(C)** Representative pictures of primary SOD1^G93A rat motor neurons treated with glutamate 5 $\mu$M for 20 min in presence of DMSO (vehicle) or Sephin1 50 nM 24 h after glutamate intoxication. Motor neurons are labeled with the neuronal marker, MAP-2. Right panel: enlargement of motor neurons **(D)** Cleaved caspase 3 level measured 24 h after glutamate intoxication in primary WT rat motor neurons culture treated with glutamate 5 $\mu$M for 20 min in presence of DMSO (vehicle) or Sephin1 50 nM. **(E)** Cell viability in glutamate intoxicated primary WT rat motor neurons treated for 24 h with DMSO (Vehicle), Sephin1, or riluzole. **(F)** Total neurite length of primary WT rat motor neurons treated for 24 h with DMSO (Vehicle), Sephin1 or riluzole 24 h after glutamate intoxication. **(G)** Cleaved caspase 3 level

# Results

The efficacy of the ER stress modulator, Sephin1, to protect motor neurons and modulate TDP-43 localization and its toxicity has been first evaluated in two in vitro models, glutamate intoxicated motor neurons and arsenite intoxicated human neuroblastoma cell line, SH-SY5Y, and then in two genetic ALS animal models, SOD1$^{G93A}$ transgenic mice and TDP-43$^{G348C}$ transgenic zebrafish embryos.

### Sephin1 protects motor neurons against glutamate excitotoxicity

To evaluate if Sephin1 can protect motor neurons against glutamate excitotoxicity, primary motor neurons from WT rats are treated 1 h before the addition of glutamate 5 μM with vehicle (DMSO), Sephin1, or riluzole. Vehicle, Sephin1, or riluzole are present during the 20 min of glutamate intoxication and during the 24 h following glutamate intoxication. Cell viability and total neurite length are evaluated by immunolocalization 24 h after glutamate intoxication by measuring the number of cells and the length of their neurites (Fig 1A and B). In this in vitro glutamate excitotoxicity assay on primary WT motor neurons, glutamate 5 μM applied for 20 min induces, 24 h later, a trend to increase cleaved caspase 3 level in primary WT rat motor neurons enriched cultures, whereas in primary WT rat motor neurons, the cell viability is significantly decreased by 35% (Figs 1B and D and S1A and B). The total neurite length is significantly reduced by 37% (Fig S1C). Riluzole 5 μM, an anti-glutamatergic agent, improves motor neuron viability by 28% and the neurite length by 45% (Fig 1E and F) compared with vehicle (0 nM Sephin1) treated motor neurons. Sephin1 treatment at 50 nM tends to reduce cleaved caspase 3 expression level (Fig 1D). Whereas 100 nM of Sephin1 tends to increase cell viability, the treatment with Sephin1 at 500 nM significantly improves by 26% the cell viability 24 h after glutamate intoxication (Fig 1B and E). However, Sephin1 treatment does not increase the total neurite length in glutamate intoxicated primary WT motor neurons (Fig 1B and F).

To evaluate if Sephin1 could protect SOD1$^{G93A}$ motor neurons against glutamate excitotoxicity, primary rat SOD1$^{G93A}$ motor neurons are treated with vehicle, Sephin1 or riluzole, 1 h before glutamate intoxication, during the 20 min of glutamate intoxication and then during the 24 h after glutamate intoxication. Motor neuron viability and total neurite length are evaluated 24 h later (Fig 1A). Glutamate intoxication in primary rat SOD1$^{G93A}$ motor neurons enriched cultures provokes a trend to increase cleaved caspase 3 protein level and a significant 40% reduction in cell viability (Figs 1C and G and S1E). The total neurite length decreases by 37% 24 h after glutamate intoxication (Figs 1C and S1F). In primary SOD1$^{G93A}$ motor neurons, 5 μM riluzole improves motor neuron cell viability by 31% and increases total neurite length by 46% compared with glutamate

intoxicated motor neurons treated with vehicle (0 nM Sephin1) (Fig 1H and I). Sephin1 treatment at 50 nM shows a trend toward a reduction of cleaved caspase 3 expression level after glutamate intoxication in primary SOD1$^{G93A}$ motor neurons enriched culture (Fig 1G). Sephin1 treatment, ranging from 10 to 500 nM, improves primary SOD1$^{G93A}$ motor neuron viability in the same range as riluzole 5 μM (Fig 1C and H) with a significant effect observed at 50 nM (32% increase). Sephin1 increases total neurite length by more than 30% with a significant effect observed at 10 nM (Fig 1C and I).

At concentrations below 500 nM, Sephin1 improves the survival of glutamate intoxicated primary motor neurons from WT and SOD1$^{G93A}$ rats. This increase of survival is accompanied by a protection of the neurite network in SOD1$^{G93A}$ rat motor neurons.

### Sephin1 reduces mitochondrial ROS without modulating calcium influx and eIF2α phosphorylation level following glutamate intoxication in primary motor neurons

To determine whether Sephin1 could improve survival by reducing calcium flux, primary WT rat motor neurons enriched cultures are treated 4 h before the addition of glutamate 5 μM with vehicle, Sephin1, or riluzole. Calcium influx is measured by the fluorescence of Fluo 4AM for 5 min after the addition of glutamate. In primary WT motor neurons enriched culture, the addition of glutamate induces a significant increase of calcium influx (Fig 2A). Riluzole (5 μM) added 4 h before the stimulation with glutamate reduces calcium influx to control level (Fig 2A). The calcium influx was not modulated by Sephin1 at 100 nM or 500 nM added 4 h before glutamate stimulation in primary WT rat motor neurons compared with vehicle treated motor neurons (Fig 2A).

To estimate the level of phosphorylation of eIF2α following glutamate intoxication and Sephin1 treatment, primary motor neurons from WT rats and SOD1$^{G93A}$ rats are treated by Sephin1 50 nM 1 h before glutamate 5 μM intoxication, then during and after the 20 min of glutamate intoxication. The level of eIF2α phosphorylation was estimated 24 h after the 20 min of glutamate intoxication (Fig 1A) by Western blot. Glutamate intoxication significantly reduces, 24 h later, the level of eIF2α phosphorylation in primary rat WT and SOD1$^{G93A}$ motor neurons enriched cultures (Fig 2B–E). Sephin1 at 50 nM, a concentration which significantly increases cell survival in primary rat SOD1$^{G93A}$ motor neurons, does not modulate the level of eIF2α phosphorylation 24 h after glutamate intoxication in WT and SOD1$^{G93A}$ rat motor neurons intoxicated by glutamate (Fig 2B–E).

To evaluate whether mitochondrial ROS is reduced by Sephin1 treatment, primary SOD1$^{G93A}$ rat motor neurons are incubated 1 h before the addition of glutamate 5 μM with Sephin1, during the 20 min of incubation with glutamate and after glutamate intoxication (Fig 1A). The level of mitochondrial ROS is evaluated by immunolocalization 4 h after glutamate intoxication by using the

measured 24 h after glutamate intoxication in primary SOD1$^{G93A}$ rat motor neurons culture treated with glutamate 5 μM for 20 min in presence of DMSO (vehicle) or Sephin1 50 nM 24 h. **(H)** Cell viability in glutamate intoxicated primary SOD1$^{G93A}$ rat motor neurons treated for 24 h with DMSO (Vehicle), Sephin1 or riluzole. **(I)** Total neurite length of primary WT rat motor neurons treated for 24 h with DMSO (Vehicle), Sephin1 or riluzole 24 h after glutamate intoxication. *$P < 0.05$, **$P < 0.01$ versus 0 nM Sephin1 Kruskal-Wallis test followed by Dunn's multiple comparison test; mean ± SEM n = 4–6 replicates per condition.
Source data are available for this figure.

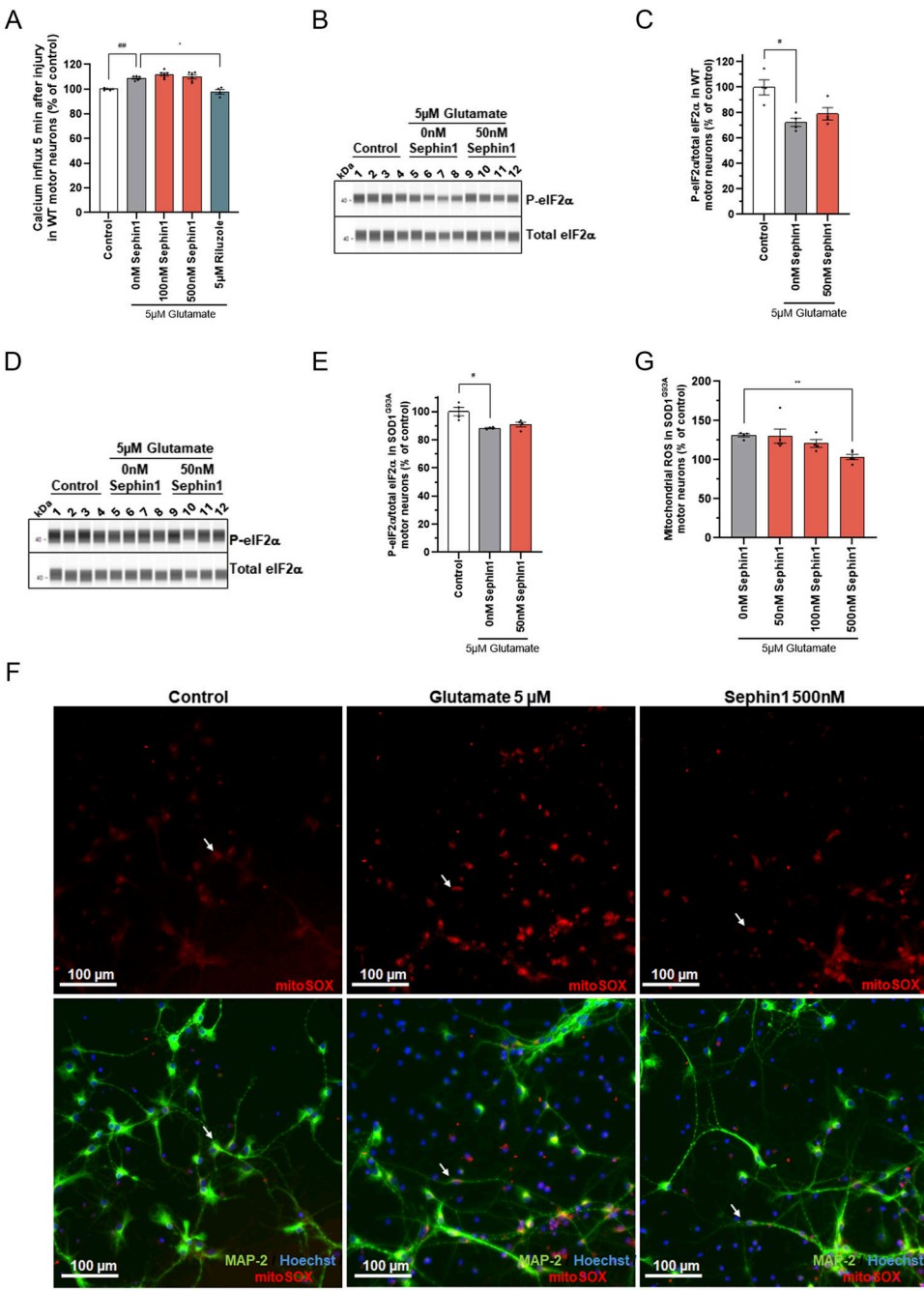

**Figure 2. Sephin1 reduces mitochondrial ROS without modulating calcium flux or the phosphorylation of eIF2α in primary rat motor neurons against glutamate intoxication.**
**(A)** Calcium influx level in primary WT rat motor neurons treated with DMSO, Sephin1 or riluzole measured 5 min after the glutamate intoxication. **(B)** Representative pictures obtained by WES automated WB apparatus of the eIF2α phosphorylated and eIF2α immunoblots in primary WT rat motor neurons intoxicated with glutamate and treated with DMSO or Sephin1 50 nM. **(C)** eIF2α phosphorylation level in primary WT rat motor neurons intoxicated with glutamate and treated with DMSO or Sephin1 50 nM 24 h after glutamate intoxication. **(D)** Representative pictures obtained by WES automated WB apparatus of the eIF2α phosphorylated and eIF2α immunoblots in primary SOD1$^{G93A}$ rat motor neurons intoxicated with glutamate and treated with DMSO or Sephin1 50 nM. **(E)** eIF2α phosphorylation level in primary SOD1$^{G93A}$ rat motor neurons intoxicated with glutamate and treated with DMSO or Sephin1 24 h after glutamate intoxication. **(F)** Representative pictures of mitoSOX staining (top panel) in primary SOD1$^{G93A}$ rat motor neurons enriched culture intoxicated with glutamate and treated with DMSO or Sephin1 500 nM, 4 h after glutamate intoxication. Representative images of mitoSOX staining overlapped with MAP-2 staining and Hoechst (bottom panel) in primary SOD1$^{G93A}$ rat motor neurons enriched culture intoxicated with glutamate and treated with DMSO or Sephin1 500 nM, 4 h after glutamate intoxication. White arrow: mitoSOX staining in MAP-2 labeled motor neurons. **(G)** Mitochondrial ROS level in primary SOD1$^{G93A}$ rat motor neurons intoxicated with glutamate and treated with DMSO or Sephin1. $^{#}P < 0.05$, $^{##}P < 0.01$ versus 0 nM Sephin1 Mann-Whitney test; $^{**}P < 0.01$ versus 0 nM Sephin1 Kruskal-Wallis test followed by Dunn's multiple comparison test; mean ± SEM n = 4–6 replicates per condition.
Source data are available for this figure.

mitochondria specific dye MitoSOX which becomes fluorescent upon its oxidation by superoxide (Fig 2F). In primary SOD1$^{G93A}$ motor neurons, a 30% increase of mitochondrial ROS is observed 4 h after the removal of glutamate (Figs 2F and S1H). Whereas Sephin1 at 100 nM tends to reduce mitochondrial ROS, 500 nM of Sephin1 significantly reduces mitochondrial ROS in these cells after glutamate intoxication (Fig 2F and G).

Following glutamate intoxication, Sephin1, in this setting, reduces mitochondrial ROS without modulating calcium influx or increasing eIF2α phosphorylation.

## Sephin1 reduces extranuclear TDP-43 in glutamate intoxicated primary motor neurons

To evaluate whether Sephin1 could reduce TDP-43 extranuclear mislocalization following glutamate excitotoxicity, primary motor neurons from WT or SOD1$^{G93A}$ rat are treated with Sephin1 or riluzole 1 h before glutamate intoxication, during and then after the 20 min of glutamate intoxication (Fig 1A). The extranuclear localization of TDP-43 is estimated by immunolocalization 24 h after glutamate intoxication. The area overlapping TDP-43 and MAP-2 immunostaining and

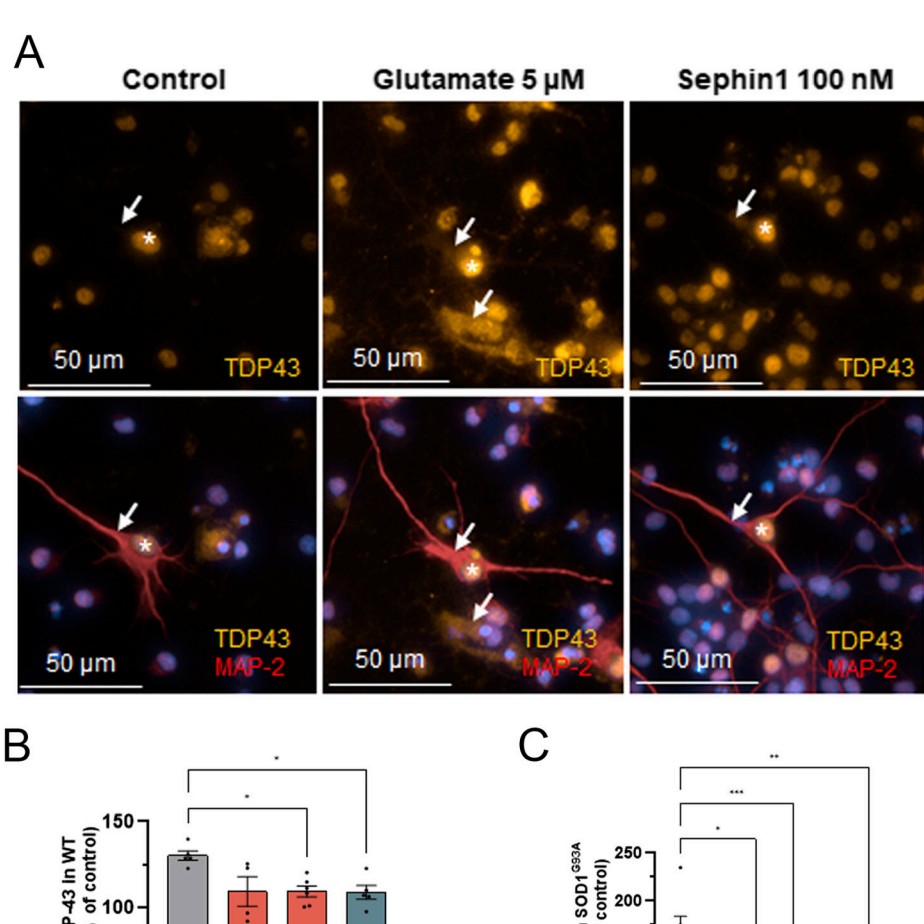

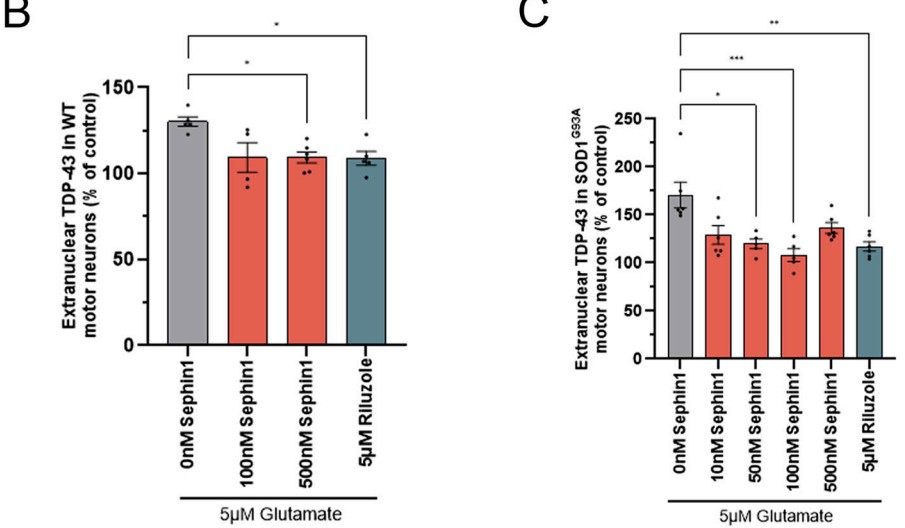

Figure 3. Sephin1 reduces extranuclear TDP-43 localization in primary motor neurons from WT and SOD1[G93A] rats.
**(A)** Representative pictures of TDP-43 localization in control primary SOD1[G93A] rat motor neurons, in primary SOD1[G93A] rat motor neurons intoxicated by glutamate and in primary SOD1[G93A] rat motor neurons intoxicated by glutamate in presence of Sephin1 100 nM. White arrows: TDP-43 localization in cytoplasm of MAP-2 labeled motor neurons, white star: TDP-43 localization in nucleus of MAP-2 labeled motor neurons. **(B)** Extranuclear TDP-43 level in primary WT rat motor neurons intoxicated by glutamate and treated with DMSO, Sephin1, or riluzole 5 $\mu$M for 24 h. **(C)** Extranuclear TDP-43 level in primary SOD1[G93A] rat motor neurons intoxicated by glutamate and treated with DMSO, Sephin1, or riluzole 5 $\mu$M for 24 h *$P < 0.05$, **$P < 0.01$, ***$P < 0.001$ versus 0 nM Sephin1 Kruskal-Wallis test followed by Dunn's multiple comparison test; mean ± SEM n = 4–6 replicates per condition.

excluding the area overlapping Hoechst (nucleus) and TDP-43 staining is considered as TDP-43 extranuclear localization. The signal of extranuclear TDP-43 could then be diffused or more pronounced like in aggregates (Fig 3A). The extranuclear TDP-43 measurement does not take into account whether TDP-43 is present in aggregates or in stress granules in the cytoplasm. In primary motor neurons from WT or SOD1[G93A] rats, 20 min of glutamate intoxication increases the extranuclear localization of TDP-43, 24 h later (Figs 3A and S1D and G). The increase of extranuclear TDP-43 localization is higher in primary SOD1[G93A] rat motor neurons than in primary WT rat motor neurons, 70% and 30%, respectively (Fig S1D and G). The anti-glutamatergic agent, riluzole (5 $\mu$M), improves cell viability in primary WT and SOD1[G93A] motor neurons and decreases extranuclear TDP-43 localization in

surviving motor neurons. The reduction of extranuclear TDP-43 localization by riluzole is higher in primary SOD1[G93A] rat motor neurons than in primary WT rat motor neurons (31% and 16% respectively) (Fig 3B and C). In WT primary motor neurons, Sephin1 at 500 nM significantly reduces extranuclear TDP-43 by 16%, to the same extent as riluzole 5 $\mu$M (Fig 3B). In primary SOD1[G93A] rat motor neurons, Sephin1 at 50 and 100 nM significantly reduces the extranuclear TDP-43 localization, 24 h after glutamate intoxication, by 30% and 46%, respectively (Fig 3A and C). The reduction of extranuclear TDP-43 localization by Sephin1 at 100 nM is higher than by riluzole 5 $\mu$M (46% versus 31%) (Fig 3C).

Sephin1 and riluzole reduce extranuclear TDP-43 localization in primary motor neurons from WT and SOD1[G93A] rats following glutamate intoxication. The decrease of extranuclear TDP-43 by

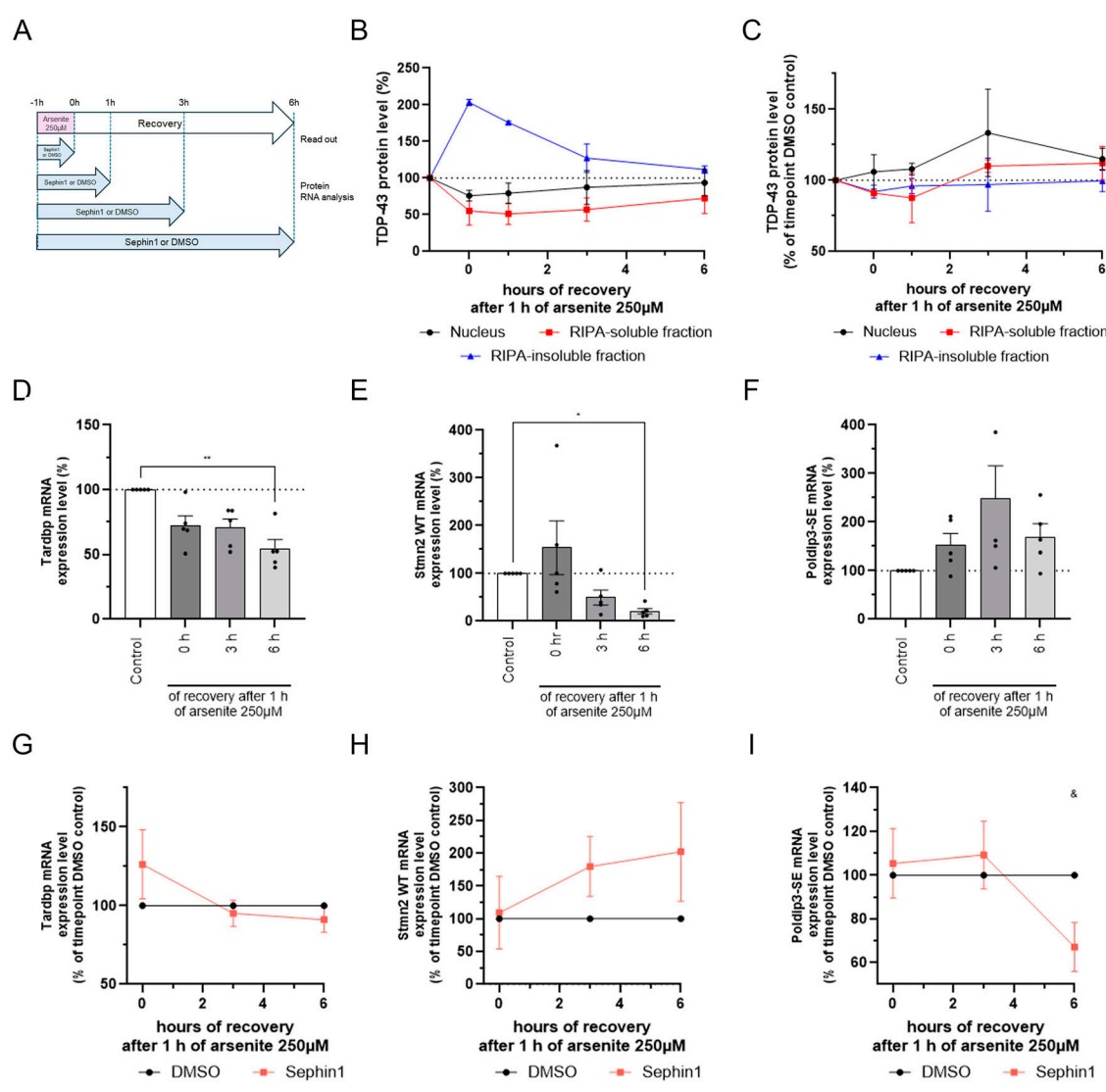

**Figure 4. Sephin1 reduces abnormal splicing during the recovery period after 1 h of arsenite stress.**
**(A)** Schematic of the experiment on SH-SY5Y. **(B)** Expression level of TDP-43 in nuclear fraction, RIPA-soluble fraction and RIPA-insoluble (urea) fraction during the recovery period after 1 h of arsenite treatment (n = 3 independent experiments). **(C)** Expression level of TDP-43 in nuclear fraction, RIPA-soluble fraction and RIPA-insoluble fraction during the recovery period after 1 h of arsenite treatment in SH-SY5Y treated with Sephin1 10 $\mu$M compared with SH-SY5Y treated with DMSO (n = 3 independent experiments). **(D)** Tardbp mRNA expression level in arsenite treated SH-SH5Y at 0, 3, and 6 h of recovery (n = 5 independent experiments). **(E)** Stmn2 WT mRNA expression level in arsenite treated SH-SY5Y at 0, 3, and 6 h of recovery (n = 5 independent experiments). **(F)** Poldip3-SE mRNA expression level in arsenite treated SH-SH5Y at 0, 3, and 6 h of recovery (n = 5 independent experiments). **(G)** Tardbp mRNA expression level at 0, 3, and 6 h of recovery in arsenite stressed SH-SY5Y treated with Sephin1 10 $\mu$M compared with SH-SY5Y treated with DMSO (n = 5 independent experiments). **(H)** Stmn2 WT mRNA expression level at 0, 3, and 6 h of recovery in arsenite stressed SH-SY5Y treated with Sephin1 10 $\mu$M compared with SH-SY5Y treated with DMSO (n = 5 independent experiments). **(I)** Poldip3-SE mRNA expression level at 0, 3 and 6 h of recovery in arsenite stressed SH-SY5Y treated with Sephin1 10 $\mu$M compared with SH-SY5Y treated with DMSO (n = 5 independent experiments). *$P < 0.05$, **$P < 0.01$ Friedman test followed by Dunn's multiple comparison test; &$P < 0.05$ two-way ANOVA followed by Sidak's multiple comparison test. Mean ± SEM.

Sephin1 is higher in primary SOD1[G93A] motor neurons compared with primary WT motor neurons.

## Sephin 1 reduces abnormal splicing during stress recovery

We interrogate if Sephin1 could reduce abnormal splicing following stress by reducing TDP-43 extranuclear localization. The human neuroblastoma cell line, SH-SY5Y, is stressed with sodium arsenite

250 $\mu$M. After 1 h, sodium arsenite medium is replaced by fresh medium for 1, 3, and 6 h. Sephin1 10 $\mu$M or vehicle (DMSO) is present during and after arsenite intoxication (Fig 4A).

TDP-43 expression level was estimated for each time point in the nuclear enriched protein fraction, RIPA-soluble protein fraction and in the RIPA-insoluble (urea) fraction. The treatment of SH-SY5Y cells with arsenite for 1 h provokes a trend toward a reduction of TDP-43 protein level in the nuclear enriched fraction and in RIPA-soluble fraction and

a significant increase of TDP-43 protein level in the RIPA-insoluble fraction (Figs 4B and S2A, B, D, E, G, and H). During the recovery period, TDP-43 protein level decreases in the RIPA-insoluble fraction (Figs 4B and S2H) whereas it tends to increase in the nuclear enriched fraction (Figs 4B and S2B). After 6 h of recovery, TDP-43 protein level in nuclear fraction, RIPA-soluble fraction and RIPA-insoluble fraction have almost returned to control level (Figs 4B and S2B, E, and H). Altogether, 1 h of arsenite treatment leads to the translocation of TDP-43 from the nucleus to the RIPA-insoluble fraction. During the recovery period, TDP-43 present in RIPA-insoluble fraction returns in less than 6 h in the nucleus (Fig 4B). The presence of Sephin1 10 $\mu$M during the arsenite treatment and the recovery period tends to increase TDP-43 in the nuclear fraction (Figs 4C and S2C). In RIPA-soluble fraction, Sephin1 significantly reduces TDP-43 level at the beginning of the recovery period and then the level of TDP-43 tends to increase at 3 and 6 h of recovery (Figs 4C and S2F). In RIPA-insoluble fraction, the protein level of TDP-43 tends toward a reduction at the beginning of the recovery period and then returns to DMSO treated level at 3 and 6 h of recovery (Figs 4C and S2I). In presence of Sephin1 10 $\mu$M, less TDP-43 proteins translocate from the nucleus into the RIPA-insoluble fraction following arsenite treatment (Fig 4C).

To evaluate whether Sephin1 could reduce abnormal splicing due to nuclear TDP-43 loss of function, the mRNA expression level of TDP-43 (Tardbp), Stathmin-2 WT (Stmn2), exon-skipping form of POLDIP3 (Poldip3-SE) were evaluated during the recovery period following arsenite stress (Fig 4A). Arsenite treatment reduces Tardbp mRNA level during the recovery period reaching a significant decrease at 6 h of recovery (Fig 4D). Stmn2 WT mRNA level decreases during the recovery period reaching a significant decrease at 6 h of recovery (Fig 4E). At the protein level, STMN2 is significantly decreased after 3 h of recovery and then increased after 6 h of recovery (Fig S2J and K). The mRNA expression level of Poldip3 lacking exon 3 tends to increase during the recovery period (Fig 4F). Arsenite treatment induces abnormal splicing during arsenite recovery period.

In presence of Sephin1 10 $\mu$M, Tardbp mRNA level is not significantly modulated during the recovery period (Fig 4G). Stmn2 WT mRNA level is increased during the recovery period by Sephin1 treatment (Fig 4H). At protein level, STMN2 protein level is higher in Sephin1-treated cells at 3 h of recovery (Fig S2K). Sephin1 10 $\mu$M reduces significantly the Poldip3-SE mRNA level at 6 h (Fig 4I). During the recovery period after arsenite intoxication, Sephin1 treatment prevents WT Stmn2 mRNA reduction and abnormal splicing. Altogether, Sephin1 treatment slightly reduces the translocation of TDP-43 to the RIPA-insoluble fraction and reduces the abnormal splicing due to TDP-43 nuclear loss of function.

## Sephin1 protects motor neurons and reduces TDP-43 in TIF in the spinal cord of SOD1[G93A] mouse model

We show that Sephin1 could reduce TDP-43 mislocalization in primary SOD1[G93A] rat motor neurons, and we interrogate whether Sephin1 could reduce TDP-43 cytoplasmic localization and/or aggregates and could improve motor neuron survival in the SOD1[G93A] mouse model. These SOD1[G93A] mice present signs of ER stress, abnormal glutamate homeostasis, cytoplasmic TDP-43 mislocalization, and an increase of TDP-43 in TIF of spinal cord

(Milanese et al, 2011; Das et al, 2015; Marino et al, 2015; Bonifacino et al, 2016; Jeon et al, 2019). In a previous study, Sephin1, administered once a day at 5 mg/kg from the age of 28 d, improved motor neuron survival in the spinal cord of SOD1[G93A] mice (Das et al, 2015). In the present study, following the guidelines for preclinical animal research in ALS/MND (Ludolph et al, 2010), Sephin1 is orally administered by oral gavage once a day at 4 or 8 mg/kg to the ALS model SOD1[G93A] mice from an early pathological stage, in absence of detectable muscle force deficit (8 wk old). At this age, motor neurons and other cellular defects are observed, such as mitochondrial alterations (Bendotti et al, 2001a; Xiao et al, 2018) and a decrease of intraepidermal nerve fiber density (Sassone et al, 2016). Histological analysis, protein expression and NFL plasma levels were evaluated in 20-wk old female mice treated daily for 12 wk by vehicle, Sephin1 4 or 8 mg/kg (Fig 5A). Motor function, as assessed by grip strength and rotarod performance was measured during the 8 wk of treatment. Survival was estimated on mice treated daily with vehicle, Sephin1 4 or 8 mg/kg from 8 wk of age.

At the age of 20 wk, female SOD1[G93A] mice exhibit a 73% reduction in the number of motor neurons in the lumbar spinal cord compared with WT mice (Fig 5B and C). This effect is associated with a significant increase of the plasma NFL level in SOD1[G93A] mice versus WT mice (Fig 5E).

After 12 wk of Sephin1 treatment, beginning at 8 wk of age, motor neuron survival increases in a dose-dependent manner with a significant improvement observed at 8 mg/kg compared with vehicle-treated SOD1[G93A] mice (Fig 5D). This enhanced motor neuron survival is accompanied by a downward trend in plasma NFL levels (Fig 5E).

After 12 wk of treatment, Sephin1 dose-dependently elevates eIF2$\alpha$ phosphorylation in the lumbar spinal cord, with a significant effect at the 8 mg/kg dose (Fig 6A and B).

Daily administration of Sephin1 at either 4 mg/kg or at 8 mg/kg, starting at 8 wk of age, does not reduce mutant human SOD1 levels, in the spinal cord protein extracts from SOD1[G93A] mice after 12 wk of treatment (Fig 6C and D).

Since TDP-43 has been shown to accumulate in the TIF of the spinal cord in female SOD1[G93A] mice (Marino et al, 2015), the level of this protein is evaluated in the TIF of the thoracic-cervical spinal cord tract of mice treated with Sephin1 or vehicle. At 20 wk, Sephin1 reduces the levels of insoluble TDP-43 protein by about 36% and 40% at 4 and 8 mg/kg ($P$ = 0.0952 vehicle versus Sephin1 8 mg/kg, Mann-Whitney test), in SOD1[G93A] mice (Fig 6E and F).

Although Sephin1 treatment at 8 mg/kg significantly enhances motor neuronal survival, it does not lead to an improvement in motor function, as assessed by grip strength and rotarod performance (Fig S3B and C). In addition, there is no improvement in body weight, disease onset or overall survival (Fig S3A, D, and E).

Thus, despite the lack of impact on motor function or survival, Sephin1 treatment is effective in reducing motor neuron loss and TDP-43 proteinopathy after 12 wk of treatment.

## Sephin1 improves survival and motor function in mutated TDP-43 transgenic zebrafish

So far, the impact of Sephin1 on motor neuron survival has been evaluated exclusively in SOD1[G93A] rodent models. Like SOD1[G93A]

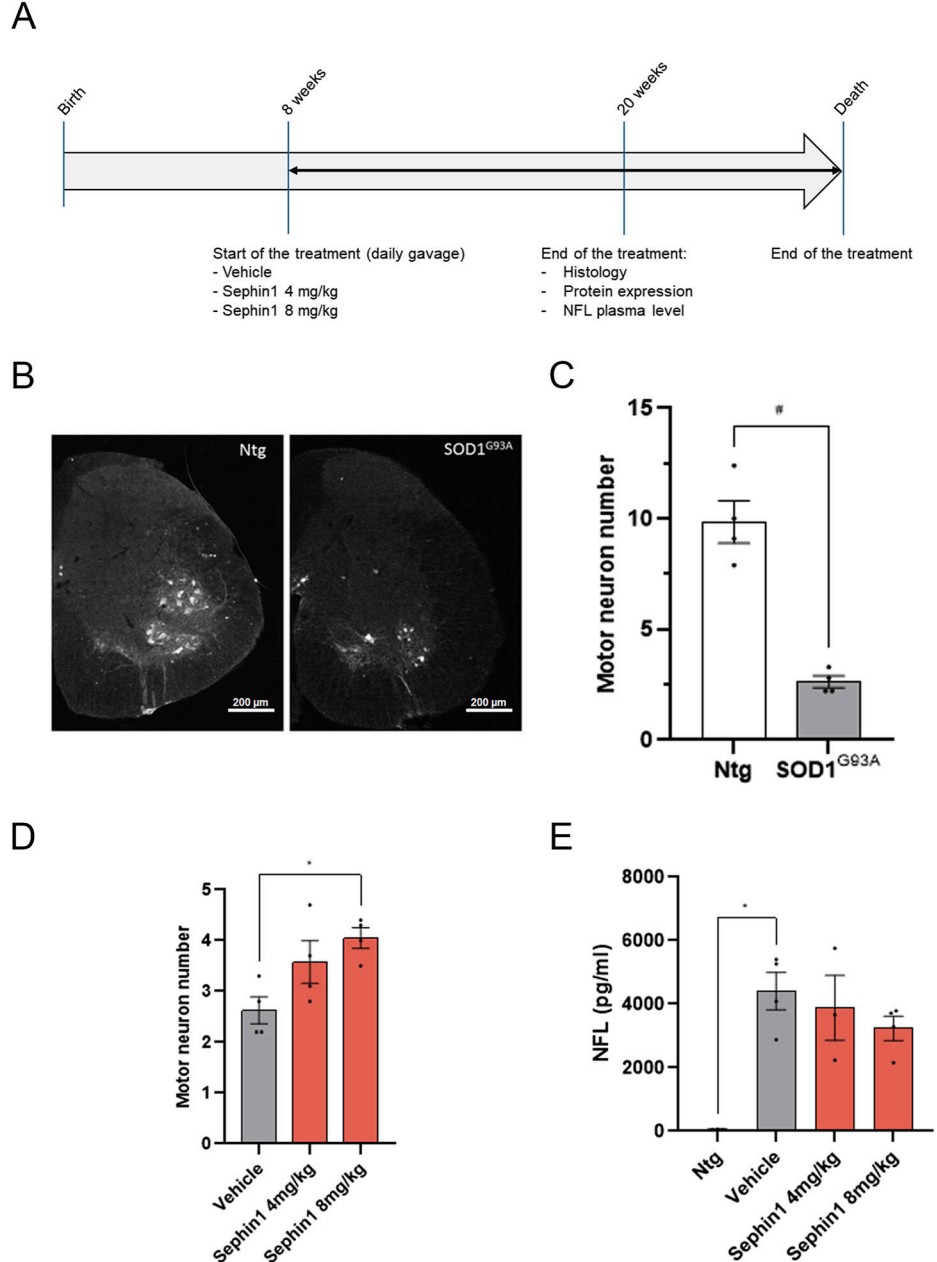

**Figure 5. Sephin1 improves survival of spinal cord motor neurons from SOD1$^{G93A}$ female mice at 20 wk of age.**
**(A)** Schematic of the experiment on female SOD1$^{G93A}$ mice. **(B)** Representative pictures of motor neurons labeled with ChAT antibody in hemisection of spinal cord of WT (Ntg) and SOD1$^{G93A}$ mice at 20 wk of age (magnification 4×). **(C)** Motor neuron number in spinal cord of 20 wk-old WT or SOD1$^{G93A}$ mice. **(D)** Motor neuron number in spinal cord of 20-wk-old SOD1$^{G93A}$ mice treated with vehicle, Sephin1 at 4 or 8 mg/kg per day. **(E)** Plasmatic NFL level in 20-wk-old WT or SOD1$^{G93A}$ mice treated with vehicle, Sephin1 at 4 or 8 mg/kg per day. $^{#}P < 0.05$ Mann-Whitney test; $^{*}P < 0.05$ versus vehicle Kruskal-Wallis test followed by Dunn's multiple comparison test; mean ± SEM n = 3–5 mice per condition.

mice, the mutated TDP-43 transgenic zebrafish model shows ER stress (Vaccaro et al, 2013). To determine whether Sephin1 can protect motor neurons against TDP-43 toxicity, the effect of Sephin1 is evaluated in the mutated TDP-43$^{G348C}$ transgenic zebrafish model. This zebrafish model stably expresses mutant TDP-43$^{G348C}$ protein following two heat shocks, the first one applied at 32 h post fertilization (hpf) and the second one at 48 hpf. The stable expression of TDP-43$^{G348C}$ protein upon heat shock is associated with locomotor defects and motor neuron axonopathy (Lissouba et al, 2018).

To evaluate the impact of Sephin1 on motor neuron survival and locomotor defects, Sephin1 10 $\mu$M or vehicle (DMSO) is applied to zebrafish embryos immediately after the first heat shock at 32 hpf.

The treatment continues for 18–20 h until motor function assessment, immunohistology analysis, and TDP-43 expression level assessment at 50–51 hpf (Fig 7A). Sephin1 has been described to increase the phosphorylation of eIF2$\alpha$ which leads to an attenuation of protein translation. To evaluate if Sephin1 reduces TDP-43 expression in zebrafish embryos, the level of TDP-43 expression is visualized by Western blot. Sephin1 10 $\mu$M treatment of zebrafish embryos does not modulate TDP-43 expression detected by a C-terminal TDP-43 antibody. However, this antibody detects a band at ~75 kD only in the zebrafish embryos in which two heat shocks had been applied to induce the expression of the TDP-43$^{G348C}$ protein. Sephin1 does not modulate the intensity of this band either (Fig S4A).

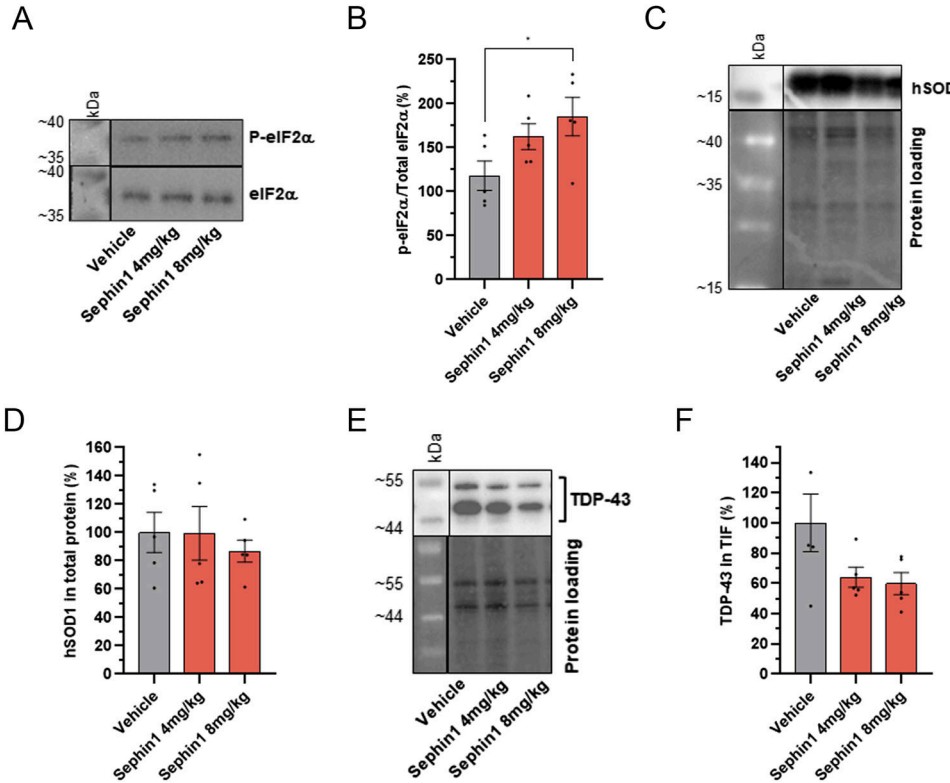

**Figure 6. Sephin1 increases eIF2α phosphorylation level and reduces TDP-43 in TIF from spinal cord motor neurons of SOD1$^{G93A}$ female mice at 20 wk of age.**
**(A)** Representative pictures of immunoblot of eIF2α phosphorylated and eIF2α proteins in spinal cord of 20-wk-old SOD1$^{G93A}$ mice treated with vehicle, Sephin1 at 4 or 8 mg/kg per day. **(B)** eIF2α phosphorylation level in the spinal cord of 20-wk-old SOD1$^{G93A}$ mice treated with vehicle, Sephin1 at 4 or 8 mg/kg per day. **(C)** Representative images of immunoblot of hSOD1 protein in spinal cord of 20-wk-old SOD1$^{G93A}$ mice treated with vehicle, Sephin1 at 4 or 8 mg/kg per day. **(D)** hSOD1 protein level in spinal cord of 20-wk-old SOD1$^{G93A}$ mice treated with vehicle, Sephin1 at 4 or 8 mg/kg per day. **(E)** Representative picture of TDP-43 immunoblot of TIF from spinal cord of 20-wk-old SOD1$^{G93A}$ mice treated with vehicle, Sephin1 at 4 or 8 mg/kg per day. **(F)** TDP-43 level in TIF from spinal cord of 20-wk-old SOD1$^{G93A}$ mice treated with vehicle, Sephin1 at 4 or 8 mg/kg per day. *$P < 0.05$ versus vehicle Kruskal-Wallis test followed by Dunn's multiple comparison test; mean ± SEM n = 3–5 mice per condition.
Source data are available for this figure.

The stable expression of mutant TDP-43$^{G348C}$ protein reduces zebrafish embryo survival observed at 50 hpf compared with zebrafish embryos who do not express this mutated TDP-43 protein (Fig S4B). Of these zebrafish embryo survivors expressing TDP-43$^{G348C}$ protein, the number of zebrafish embryos showing locomotor defects is significantly increased compared with mTDP-43 zebrafish embryos or to WT zebrafish embryos subjected to heat shock (Fig S4C). This locomotor defect is observed as a reduction of swim distance, swim duration, and swim velocity (Fig S4D–F) measured after lightly touching the tail of 50–51 hpf embryos with blunt forceps. The stable expression of TDP-43$^{G348C}$ protein leads to a reduction of survival and an increase of locomotor dysfunction of surviving zebrafish embryos at 50–51 hpf.

The treatment with Sephin1 10 μM from 32 hpf significantly improves the survival of zebrafish embryos expressing mutant TDP-43$^{G348C}$ protein by 41% (Fig 7B). This increase in survival is associated with a 48% decrease in locomotor dysfunction (Fig 7C). The zebrafish embryos expressing TDP-43$^{G348C}$ protein treated with Sephin1 swim a longer distance and duration than those treated with vehicle (Fig 7D and E). The swim velocity of the Sephin1 treated zebrafish embryos expressing TDP-43$^{G348C}$ protein is also significantly improved (Fig 7F).

To determine whether the improvement of locomotor function by Sephin1 treatment is associated with an increase in motor neuron survival and an increase of axonal length, the number of motor neurons and the length of their motor neuron axons are measured in zebrafish embryos treated with Sephin1 or vehicle from 32 to 51 hpf. The expression of TDP-43$^{G348C}$ protein induces a significant reduction of motor neurons and a significant reduction of the axonal length of

these motor neurons (Fig S5A and B). Motor neuron survival is higher in the Sephin1-treated group ($P$ = 0.0857, Mann-Whitney test, Fig 8A and B). The axonal length of these motor neurons is also significantly increased by Sephin1 treatment (Fig 8A and C).

Sephin1 treatment improves survival and locomotor function of the zebrafish embryos expressing TDP-43$^{G348C}$ protein. The improvement of locomotor function is associated with an increase in motor neuron survival and an increase in the axonal length of these motor neurons.

## Discussion

Sephin1, also named icerguastat or IFB-088, which has completed a phase 2 clinical trial in bulbar-onset ALS patients (NCT05508074), has been previously evaluated in SOD1$^{G93A}$ mice showing beneficial effects, including improvement of motor neuron survival and decrease of SOD1 aggregates (Das et al, 2015; Vieira et al, 2024). However, these SOD1 aggregates are found only in SOD1 mutated ALS patients who account for only 1–6% of the overall ALS population (Benatar et al, 2025). The main protein found in aggregates in ALS patients is TDP-43 which is observed in 97% of ALS patients. The impact of Sephin1 on TDP-43 and on its cellular localization has not been evaluated yet. In this study, we demonstrate that Sephin1 increases the survival of primary WT and SOD1$^{G93A}$ motor neurons following glutamate intoxication. We show that Sephin1 reduces mitochondrial ROS and TDP-43 cytoplasmic mislocalization in an eIF2α

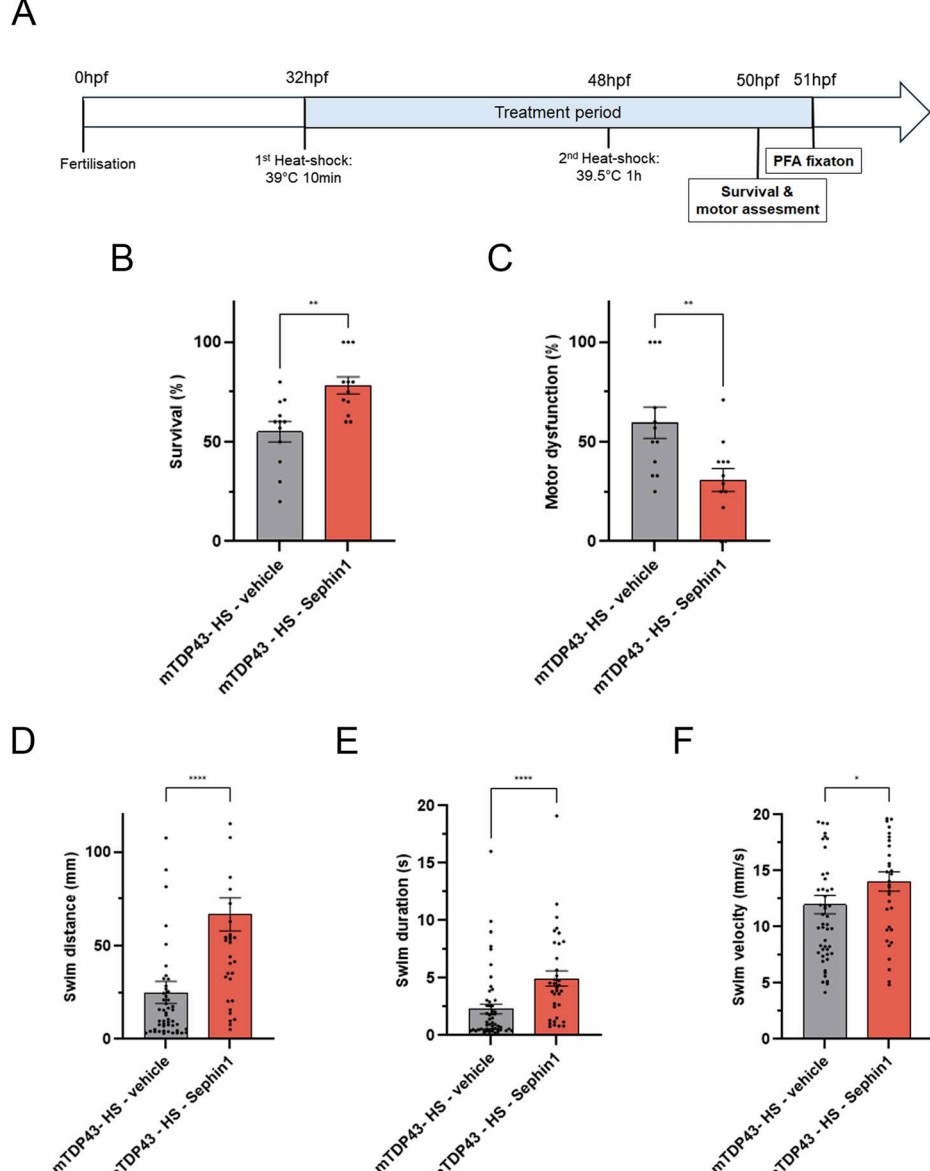

**Figure 7. Sephin1 improves survival and locomotor function in zebrafish embryos expressing mutant TDP-43$^{G348C}$ protein.**
**(A)** Schematic of the experiment in mutated TDP-43 transgenic zebrafish embryos. **(B)** Survival percentage of zebrafish embryos expressing mutant TDP-43$^{G348C}$ protein treated with vehicle or Sephin1 10 μM (**$P < 0.01$ unpaired $t$ test, n = 12 replicates per condition). **(C)** Motor dysfunction percentage of zebrafish embryos expressing mutant TDP-43$^{G348C}$ protein treated with vehicle or Sephin1 10 μM (**$P < 0.01$ unpaired $t$ test, n = 12 replicates per condition). **(D)** Swim distance of zebrafish embryos expressing mutant TDP-43$^{G348C}$ protein treated with vehicle or Sephin1 10 μM following the touch of the tail (****$P < 0.0001$ Mann-Whitney test n = 35–50 zebrafish per condition). **(E)** Swim duration of zebrafish embryos expressing mutant TDP-43$^{G348C}$ protein treated with vehicle or Sephin1 10 μM following the touch of the tail (****$P < 0.0001$ Mann-Whitney test n = 35–53 zebrafish per condition). **(F)** Swim velocity of zebrafish embryos expressing mutant TDP-43$^{G348C}$ protein treated with vehicle or Sephin1 10 μM following the touch of the tail (*$P < 0.05$ Mann-Whitney test, n = 35–50 zebrafish per condition). mTDP43-HS: zebrafish embryos expressing TDP43$^{G348C}$ protein; mean ± SEM.

phosphorylation independent manner. We also show that Sephin1 could reduce abnormal splicing due to arsenite inducing TDP-43 nuclear loss of function. In SOD1$^{G93A}$ mice, the daily administration of Sephin1 from 8 wk of age for 12 wk improves motor neuron survival, tends to decrease plasmatic NFL level and TDP-43 protein in TIF of spinal cord. Although the impact of Sephin1 is positive at the cellular level, it does not improve motor function and mice survival. We also demonstrate in another ALS animal model, the mutant TDP-43 zebrafish, that Sephin1 increases motor neuron survival, improves motor function, and increases zebrafish embryo survival.

Excitotoxicity regulated by glutamate-mediated activation of the ionotropic NMDA receptors (NMDARs) and AMPA receptors (AMPARs) is one of the hallmarks of ALS (Arnold et al, 2024). Upon their activation, the increase of calcium influx leads to the activation of calcium dependent enzymes and to the perturbation of

the calcium homeostasis in the ER and in the mitochondria, leading to ER stress, mitochondrial dysfunction and an increase of ROS. Altogether, the strong increase of calcium influx induced by high concentration of glutamate leads to neuronal death. Here, we have shown that glutamate induces a reduction of cell survival 24 h after glutamate application on both WT and SOD1$^{G93A}$ motor neurons. Sephin1 increases the survival of these primary motor neurons. These data are in agreement with those obtained by Ruis and colleagues on NMDA intoxicated primary WT neurons (Ruiz et al, 2020). In both studies, the neuroprotective effect of Sephin1 is independent of the phosphorylation of eIF2α, suggesting another mechanism of action for Sephin1 in glutamate or NMDA intoxicated neuronal cells. Ruiz et al (2020) have shown that Sephin1 decreases calcium influx and calcium-dependent enzyme activity in NMDA stimulated rat cortical neurons at the concentrations used for

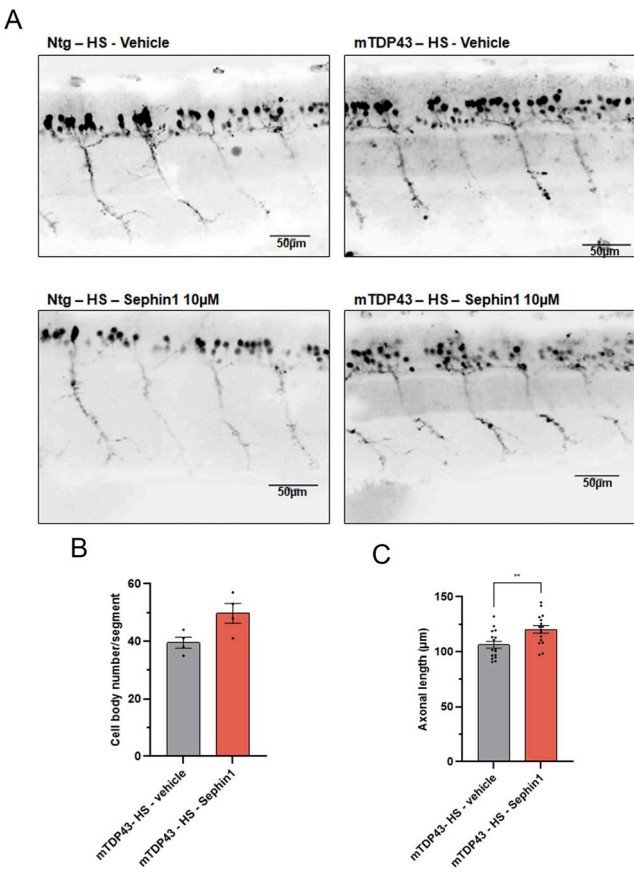

**Figure 8. Sephin1 improves motor neurons survival and axonal length in zebrafish embryos expressing mutant TDP-43^G348C protein.**
**(A)** Representative pictures of spinal cord motor neurons of non-transgenic (Ntg) or mutant TDP-43^G348C protein expressing zebrafish embryos at 51 hpf and treated with vehicle or Sephin1 10 μM. **(B)** Cell body number of motor neurons measured per segment in zebrafish embryos expressing mutant TDP-43^G348C protein treated with vehicle or Sephin1 10 μM. ($P$ = 0.0857 Mann–Whitney test, n = 4 replicates per condition). **(C)** Axonal length of motor neurons in zebrafish embryos expressing mutant TDP-43^G348C protein treated with vehicle or Sephin1 10 μM. (**$P < 0.01$ unpaired $t$ test, n = 16 motor neurons per condition). Ntg-HS: WT zebrafish with 2 heat shock, mTDP43-HS: zebrafish embryos expressing TDP43^G348C protein mean ± SEM.

neuroprotection, suggesting early NMDAR signaling through calcium influx modulation. However, in our experiment on glutamate intoxicated primary WT motor neurons, Sephin1 was not able to reduce calcium influx at the neuroprotective concentrations, suggesting a different mechanism of action independent of the calcium influx modulation and eIF2α phosphorylation. Glutamate intoxication induces an increase of mitochondrial ROS level. We showed that Sephin1 reduces mitochondrial ROS 4 h after glutamate intoxication in a dose dependent manner underlying a possible role of the mitochondria in the neuroprotective effect of Sephin1. This needs to be further investigated.

Glutamate-mediated excitotoxicity can also induce the clearance and the accumulation of nuclear TDP-43 protein into the cytoplasm (Corona et al, 2007; Boussicault et al, 2020; Obrador et al, 2020). Cytoplasmic TDP-43 localization is associated with cytotoxicity (Kabashi et al, 2010; Diaper et al, 2013; Walker et al, 2015;

Cascella et al, 2016), and the loss of TDP-43 nuclear function leads to the aberrant incorporation of cryptic exons into mRNAs resulting in aberrant protein expression and protein depletion leading to neuronal dysfunction such as axonal abnormalities (Seddighi et al, 2024). Nuclear depletion and cytoplasmic inclusion of TDP-43 are found in ALS patients (Koike, 2024). The presence of TDP-43 aggregates in motor neuron cytoplasm has also been reported in SOD1-associated ALS patients (Mackenzie et al, 2007; Tan et al, 2007) and in spinal cord of SOD1 mice model from the symptomatic stage (Okamoto et al, 2011; Jeon et al, 2019). In primary motor neurons from SOD1^G93A rats, glutamate intoxication leads to the translocation of TDP-43 from the nucleus to the cytoplasm, where it is still visible after 24 h. Sephin1 treatment at neuroprotective concentration reduces the extranuclear TDP-43 localization 24 h after glutamate intoxication in primary SOD1^G93A motor neurons. The reduction of extranuclear TDP-43 localization by Sephin1 is not dependent on the presence of mutated SOD1 in motor neurons as it is also observed in stressed primary WT motor neurons. Therefore, by reducing mitochondrial ROS levels following glutamate intoxication, Sephin1 could prevent the translocation of TDP-43 into the cytoplasm or could speed up its return into the nucleus. The precise way on how Sephin1 might reduce TDP-43 cytoplasmic mislocalization remains difficult to establish as Sephin1 was present before, during and after glutamate stress. A recent publication has described that during the recovery phase following oxidative stress, TDP-43 could undergo reversible nuclear condensates leading to TDP-43 inactivation and loss of function in splicing. These authors have shown that cytoplasmic TDP-43 localization was an early and very transient event during stress, whereas TDP-43 nucleoplasmic depletion occurred later, and was uncoupled from the cytoplasmic redistribution and persisted for longer (Huang et al, 2024). In our experiments on primary motor neurons, we cannot exclude that by preventing TDP-43 cytoplasmic localization after glutamate intoxication, Sephin1 could also reduce TDP-43 nucleoplasmic depletion and then prevent splicing loss of function leading to neuronal dysfunction. As the subset of cryptic exons expressed by nuclear TDP-43 loss of function is different between rodent and human (Ling et al, 2015), the impact of Sephin1 on abnormal splicing was evaluated in an arsenite stressed human cell line (Huang et al, 2024). Several studies have shown that TDP-43 knockdown provokes the insertion of a cryptic exon or the skipping exon in crucial mRNAs leading to neuronal dysfunction (Seddighi et al, 2024). In the case of Stathmin-2 mRNA, TDP-43 nuclear loss of function leads to the insertion of the exon 2a in the Stmn2 pre-mRNA provoking its degradation, whereas the expression of the Stmn2 WT mRNA also decreases (Melamed et al, 2019). For Poldip3 mRNA, TDP-43 nuclear loss of function leads to the skipping of the exon 3 which leads to an increase expression of this new variant (Shiga et al, 2012). The depletion of Stmn2 WT variant and the increase of skipping exon variant of Poldip3 have been observed in ALS patients (Shiga et al, 2012; Klim et al, 2019; Cao & Scotter, 2022). During the recovery following arsenite treatment, Stmn2 WT mRNA and STMN2 protein decrease, whereas the expression of skipping exon mRNA variant of Poldip3 increases. Sephin1 treatment increases the expression level of Stmn2 WT mRNA and STMN2 protein and significantly decreases the expression level of the skipping exon mRNA variant of Poldip3 during the recovery. These data suggest that Sephin1

could decrease abnormal splicing due to the TDP-43 nuclear loss of function. This decrease in the abnormal splicing could be due to an increase of TDP-43 protein expression or TDP-43 nuclear localization. This decrease in abnormal splicing by Sephin1 is unlikely due to an increase in the TDP-43 expression as the Tardbp mRNA is decreased by arsenite treatment and Sephin1 treatment does not increase its expression level during the recovery period. In the presence of Sephin1, TDP-43 protein level tends to increase in the nuclear fraction and to decrease in the RIPA soluble and insoluble fraction, suggesting that TDP-43 could be more retained in the nucleus in presence of Sephin1. Altogether, we show that Sephin1 reduces abnormal splicing following arsenite treatment likely by maintaining TDP-43 in the nucleus. Further investigation will be needed to determine how Sephin1 reduces abnormal splicing and retain TDP-43 in the nucleus.

Aberrant glutamate homeostasis leading to excitotoxicity and accumulation of TDP-43 aggregates have been described in the spinal cord of symptomatic SOD1[G93A] mice (Bendotti et al, 2001b; Marino et al, 2015). In 20 wk-old symptomatic SOD1[G93A] mice, treatment with Sephin1 at doses of 4 or 8 mg/kg for 12 wk significantly reduces TDP-43 in the TIF of spinal cord. This reduction is associated with increased motor neuron survival. These findings align with data from primary motor neurons exposed to glutamate where Sephin1 treatment improves cell survival and reduces extranuclear TDP-43. However, whereas the eIF2$\alpha$ phosphorylation level is unchanged by Sephin1 in primary motor neurons injured with glutamate, Sephin1 increases the eIF2$\alpha$ phosphorylation in a dose-dependent manner in spinal cord of SOD1[G93A] mice. It is known that eIF2$\alpha$ phosphorylation is not restricted to neurons but is also activated in microglia in response to stress conditions (Ruggieri, 2023). Recently, Wu and colleagues have been reported that in microglia, the stress-induced stress granule assembly through the phosphorylation of eIF2$\alpha$ precludes the activation of NLRP3 inflammasome, becoming neuroprotective (Wu et al, 2023). Thus, the increase of eIF2$\alpha$ phosphorylation observed in spinal cord by Sephin1 treatment may be due to the activated microglia typical of mouse and human ALS spinal cord. These data suggest that Sephin1 could also modulate ER stress response in addition to its effects on TDP-43 proteinopathy.

So far, only two studies have investigated the effects of Sephin1 treatment in SOD1[G93A] mice in relation to motor function and survival outcomes. Das et al showed that administering Sephin1 daily at 5 mg/kg by oral gavage from 4 wk of age increases motor neuron survival and improves motor function, although survival was not assessed (Das et al, 2015). Similarly, in our present study, Sephin1 treatment also significantly improves motor neuron survival. However, we did not observe an improvement in motor function, as measured by rotarod and grip strength tests. A possible explanation for the discrepancy between the two studies could be the difference in the timing of treatment initiation: 4 wk of age in Das et al (2015) versus 8 wk of age in the present study. This may be critical as Mancuso et al showed that the amplitude of the compound muscle action potential (CMAP) as well as motor evoked potentials (MEP) of the hindlimb muscles, tibialis anterior and plantar muscles of C57Bl/6 SOD1[G93A] transgenic mice, start to decline by 6 wk of age in both male and female, and further

decline up to 16 wk of age (Mancuso et al, 2012). This suggests that in our study, Sephin1 treatment was initiated after the onset of muscle defects, which could not be reversed by the treatment. Another potential explanation for the conflicting results could be the differences in the experimental protocols used to monitor disease progression. In Das et al's studies (Das et al, 2015; Luh et al, 2017), mice underwent an intense preliminary training regimen (learning) on the accelerating rotarod, potentially creating a more stressful condition that may have contributed to the accelerated decline in motor function. In contrast, our study implemented a more consistent and less demanding training and testing protocol, likely to result in milder disease progression. Recent findings suggest that intense endurance exercise can exacerbate motor dysfunction and negatively impact the disease onset of SOD1[G93A] female mice (Scaricamazza et al, 2024) supporting the notion that the more stressful condition in the Das et al study could have influenced the outcomes. This distinction may also have implications for the therapeutic response, as a more stressful condition might be more responsive to inhibitors of the stress-induced phosphatase complex, such as Sephin1, compared with a milder disease state. Future studies should explore the interaction between stress levels, disease severity, and the efficacy of stress-modulating treatments like Sephin1, in the SOD1[G93A] mice. Another publication reported that Sephin1, when administered daily intraperitoneally (ip) at 10 mg/kg, in SOD1[G93A] mice from 50 d of age, delayed disease onset by 5 d and showed a trend toward extending survival by 3 d. In contrast, a lower dose (4 mg/kg administered ip every other day) did not affect survival (Vieira et al, 2024). In the study presented here, Sephin1 treatment given daily at 4 or 8 mg/kg by oral gavage does not improve disease onset or survival. The discrepancy in outcomes may stem from differences in dosage and administration routes. The intraperitoneal route used in Viera's study likely resulted in faster and more complete absorption than oral gavage, as it is generally considered that substances absorbed from the peritoneal cavity more closely mimic intravenous administration (Al Shoyaib et al, 2019). As a result, the higher dose (10 mg/kg) given ip would have provided greater systemic exposure (area under the curve, AUC, and maximum concentration, $C_{max}$) compared with the lower 8 mg/kg dose given orally in this study. Thus, we should consider increasing the dose of Sephin1 given to mice when it is administered by oral gavage to achieve the same level of systemic exposure as the 10 mg/kg ip dose to potentially replicate the survival benefits seen in the earlier study.

To date, Sephin1 has been only evaluated in ALS rodent models involving an overexpression of mutated SOD1 protein. In the present study, the impact of Sephin1 on survival and motor function has been evaluated in zebrafish embryos expressing human TDP-43 protein harboring the mutation G348C following two heat shocks. These zebrafish embryos stably expressing TDP-43[G348C] protein show locomotor defect and motor neuron axonopathy (Lissouba et al, 2018). In our study, we show that the survival of zebrafish embryos stably expressing TDP-43[G348C] protein is reduced at 50–51 hpf compared with non-expressing TDP-43[G348C] zebrafish embryos. Sephin1 administered at 32 hpf for ~20 h improves the survival of TDP-43[G348C] expressing zebrafish embryos. In Vaccaro's study, guanabenz improves motor function and increases motor axon

length in mutated TDP-43 expressing zebrafish embryos (Vaccaro et al, 2013). Like guanabenz, Sephin1 treatment improves motor function and increases motor axon length in mutated TDP-43 expressing zebrafish embryos indicating that Sephin1 reduces mutated TDP-43 toxicity in zebrafish. In zebrafish embryos expressing TDP-43[G348C] protein, we did not observe a difference in TDP-43 protein expression level in total protein extract by Sephin1 treatment suggesting that Sephin1 does not regulate TDP-43 expression level. However, we could not exclude that Sephin1 modifies the localization of TDP-43 (endogenous or human TDP43[G348C] proteins) in motor neurons of mutated TDP-43 zebrafish embryos. The localization of mutated TDP-43 in zebrafish embryos in presence of Sephin1 will need to be investigated.

Here, we demonstrate that during glutamate excitotoxicity, Sephin1 protects motor neurons through a mechanism that is likely dependent on the reduction of mitochondrial ROS. This neuroprotective effect of Sephin1 on motor neuron survival is also observed in the spinal cord of SOD1[G93A] mice and in TDP-43[G348C] expressing zebrafish embryos in which ER stress has been described. In motor neurons, Sephin1 reduces the accumulation of TDP-43 proteins in the cytoplasm, thereby contributing to the neuroprotective action of Sephin1. It reduces abnormal splicing due to TDP-43 nuclear loss of function which could also contribute to Sephin1 neuroprotective action. In an animal model dependent on TDP-43 toxicity, Sephin1 improves motor function and prolongs the survival of mutated TDP-43 zebrafish model. Together, we demonstrated that Sephin1 could be a valuable drug candidate for the treatment of the 97% of ALS patients presenting TDP-43 pathology. Importantly, results derived from these cellular and animal models could be further extended by the evaluation of the TDP-43 protein level in plasma extracellular vesicles (EVs) from the bulbar-onset ALS patient enrolled in Phase II clinical trial (NCT05508074). EVs are small vesicles in a lipid bilayer containing protein, RNA and DNA and secreted from almost all cells and are detectable in plasma or CSF. EVs are able to cross the blood-brain barrier and are a means of intercellular signaling to and from the central nervous system (McCluskey et al, 2022). Several studies have shown that the level of TDP-43 in EVs from plasma or CSF are increased in ALS patients compared with healthy control (Sproviero et al, 2018; Vassileff et al, 2020; Chatterjee et al, 2024). The presence of TDP-43 in EVs likely reflects TDP-43 pathological relocalization from the nucleus to the cytoplasm, as nuclear export is a prerequisite for TDP-43 incorporation into EVs. The effect of Sephin1 on TDP-43 protein level will be evaluated in plasma EVs from the bulbar-onset ALS patient enrolled in Phase II clinical trial (NCT05508074). Overall, Sephin1 improves motor neuron survival in multiple preclinical ALS models by reducing TDP-43 cytoplasmic mislocalization and reducing abnormal splicing due to TDP-43 nuclear loss of function suggesting that Sephin1 could be a therapeutic strategy for ALS patients.

# Materials and Methods

## Material

Chemicals, antibodies, and primers used in this study are listed in Tables 1 and 2.

## Primary rat motor neurons experiments

### Primary culture of spinal motor neurons

All experiments were carried out in accordance with the French National Institutes of Health Guide for the Care and Use of Laboratory Animals and followed current European Union regulations (Directive 2010/63/EU). Experimental procedures were approved by the National and Institutional Ethical Committees. Rat spinal cord motor neurons (MNs) from WT (Sprague Dawley, Janvier; Janvier Labs) or SOD1[G93A] rat (Taconic Bioscience) were cultured as described (Wang et al, 2013; Boussicault et al, 2020). Briefly, fetuses (E14) were removed from the uterus of pregnant female rats of 14 d gestation. Spinal cords were removed and dissociated with a 0.05% trypsin-0.02% EDTA solution (Dutscher) for 20 min at 37°C. After dissociation, cells were then centrifuged at 515$g$ for 10 min at 4°C. The cell pellet was resuspended in a defined culture medium Neurobasal medium (Thermo Fisher Scientific) with a 2% B27 supplement solution (Thermo Fisher Scientific), L-glutamine 2 mM, 2% of penicillin/streptomycin solution, and 10 ng/ml of brain-derived neurotrophic factor (Dutscher). The cells from the same motor neurons isolation were seeded at a density of 20,000 cells per well in 96-well plates pre-coated with poly-L-lysine (Dutscher) and incubated at 5% $CO_2$.

### Genotyping of SOD1 Tg rat embryos

Female (Sprague Dawley, Janvier) were mated at Neuro-Sys (Gardanne, France) with Tg SOD1[G93A] rat (Taconic Bioscience, US). The DNA from each embryo (E14) brain was extracted with the SYBR Green Extract-N-Amp tissue PCR kit (Sigma-Aldrich). Primers specifically for the genomic human *SOD1* gene were used to detect the presence of the SOD1[G93A] transgene (Table 2). The PCR was run using the CFX96 Bio-Rad RT–PCR system. The results for each sample were compared with negative control (ultrapure water) and to the positive control (DNA from Tg SOD1[G93A] embryos).

### Treatment on motor neurons and immunolocalization

Sephin1, riluzole 5 $\mu$M and vehicle (DMSO, Sigma-Aldrich) were applied on primary motor neurons (13 d in culture, DIV13) from WT and SOD1[G93A] rat. After 1 h of incubation, glutamate was added to a final concentration of 5 $\mu$M diluted in medium, still in presence of the compounds. After 20 min, glutamate was washed out and fresh culture medium with compounds was added for an additional 24 h. After the incubation period, primary motor neurons were fixed by a cold solution of 95% ethanol and 5% acetic acid for 5 min at −20°C, then permeabilized and blocked. The cells were incubated with anti-microtubule-associated-protein 2 (MAP-2) antibody and anti-TDP-43 antibody (Table 1). These antibodies were revealed with Alexa Fluor 488 goat anti-mouse IgG and Alexa Fluor 568 goat anti-rabbit IgG antibodies (Sigma-Aldrich). Nuclei were counterstained with the fluorescent dye Hoechst (Sigma-Aldrich).

To evaluate mitochondrial ROS, the compounds, DMSO and Sephin1, were applied on primary SOD1[G93A] rat motor neurons DIV13. After 1 h of incubation, glutamate 5 $\mu$M was applied on the motor neurons still in the presence of the compounds. After 20 min, glutamate was washed out and fresh culture medium with compounds was added for an additional 4 h. Then live cells were incubated with MitoSOX Red (Thermo Fisher Scientific). After 10 min of incubation,

**Table 1. List of chemicals and antibodies used**

| Reagent | Source | Catalog number | Figure |
|---|---|---|---|
| Chemicals | | | |
| Sephin1 | InFlectis BioScience | | Figs 1, 2, 3, 4, 5, 6, 7, and 8 and S1, S2, S3, S4, and S5 |
| Riluzole | Sigma-Aldrich | R116 | Figs 1, 2, and 3 |
| L-glutamic acid monopotassium | Sigma-Aldrich | G1501 | Figs 1, 2, and 3 and S1 |
| Antibodies | | | |
| Mouse monoclonal MAP2 | Sigma-Aldrich | M4403 | Figs 1, 2, and 3 and S1 |
| Rabbit polyclonal TDP-43 | Ozyme | 3448S | Figs 3 and S1 |
| Rabbit polyclonal eIF2α | Cell Signaling | 5324S | Fig 2 |
| Rabbit polyclonal phospho-eIF2α | Thermo Fisher Scientific | 10047382 | Fig 2 |
| Anti-caspase 3 | Cell Signaling | 9662 | Figs 1 and S1 |
| Rabbit anti Stathmin2 | ProteinTech | 10586-1-AP | Figs 4 and S2 |
| Mouse anti-alpha-tubuline | Sigma-Aldrich | T5168 | Figs 4 and S2 |
| Goat polyclonal choline o-aceyltransferase (ChAT) | Millipore | SAB2500236 | Fig 5 |
| Rabbit polyclonal hSOD1 | Upstate | 07–403 | Fig 6 |
| Rabbit polyclonal eIF2α | Cell Signaling | 9722S | Fig 6 |
| Rabbit polyclonal phospho-eIF2α | Cell Signaling | 3597S | Fig 6 |
| Rabbit polyclonal TDP-43 | Proteintech | 12892-1-AP | Fig 6 |
| Rabbit polyclonal TDP-43 | Proteintech | 10782-2-AP | Figs 4 and S2 and S4 |
| Mouse monoclonal anti-actin | Thermo Fisher Scientific | MA5-11869 | Fig S3 |

**Table 2. List of primers used.**

| Primers for PCR and TaqMan probes for RT-qPCR | Source | Catalog number | Figure |
|---|---|---|---|
| TaqMan probes for RT-qPCR | | | |
| STMN2 | Thermo Fisher Scientific | Hs00975902 | Fig 4 |
| TDP-43 | Thermo Fisher Scientific | Hs00606522 | Fig 4 |
| POLDID3 SE | Thermo Fisher Scientific | Hs00737354 | Fig 4 |
| HPRT | Thermo Fisher Scientific | Hs02800695 | Fig 4 |
| Primers for PCR | | | |
| SOD1$^{G93A}$ transgene: | | | |
| Forward: 5′-CATCAGCCCTAATCCATCTGA-3′ | | | |
| Reverse: 5′-CGCGACTAACAATCAAAGTGA-3′ | | | |

the cells were fixed with a solution of 4% PFA, permeabilized and blocked. Then, the cells were incubated with anti-MAP-2 antibody and by Alexa Fluor 488 goat anti-mouse antibody (Sigma-Aldrich). Nuclei were counterstained with the fluorescent dye Hoechst.

For each condition, 30 pictures (representative of the whole well area) per well were automatically taken using the high-content imager, ImageXpress (Molecular Devices) with 20× magnification using the same acquisition parameters for all images. Image analyses were directly and automatically performed by MetaXpress software (Molecular Devices). Neuron survival was measured by counting the number of MAP-2 labeled cells per well. The total neurite length was determined by measuring the total neurite length ($\mu$m) of cells labeled with MAP-2. The extranuclear TDP-43 analysis was performed by measuring in $\mu$m$^2$ the overlapped area between MAP-2 and cytoplasmic TDP-43 staining without the overlapped area between Hoechst and TDP-43 staining. The mitochondrial ROS analysis was performed by measuring in $\mu$m$^2$ the overlapped area between MAP-2 and MitoSOX staining.

On primary WT rat motor neurons DIV13, 3 h after the Sephin1, riluzole 5 $\mu$M or vehicle (DMSO) application, Fluo 4AM (Thermo Fisher Scientific) at 4 $\mu$M was incubated on the cells for 1 h at 37°C. Then, glutamate (5 $\mu$M) was applied onto the cells. The level of intracellular Ca$^{2+}$ (on all the cells from the culture) was evaluated by measuring the fluorescence (Ex/Em: 494/508 nm) 5 min after glutamate application, using Glomax apparatus (Promega).

### Protein analysis

Primary rat motor neurons were lysed by CelLytic MT reagent (Sigma-Aldrich), then protein concentration was determined by the micro kit BCA (Pierce, Thermo Fisher Scientific). Phospho-eIF2α, eIF2α, caspase 3, and cleaved caspase 3 protein levels were analyzed on the WES automated WB apparatus following manufacturer's recommendations (WES; ProteinSimple) using rabbit anti-phospho-eIF2α, rabbit anti-eIF2α and rabbit anti-caspase 3 antibodies. Data analysis was performed using Compass Software (ProteinSimple).

## SH-SY5Y experiments

### Cell line and treatment

SH-SY5Y cell line was obtained from Merck. Cells were grown in DMEM/F12 medium (Dutscher) supplemented with 10% FBS (Dutscher), 1X nonessential amino-acids (Gibco) and 1X penicillin streptomycin (Dutscher). Cells were treated with 250 µM of sodium arsenite (Merck) for 1 h in the presence of DMSO or Sephin1 10 µM. For recovery, sodium arsenite containing medium was replaced with fresh medium containing DMSO or Sephin1 10 µM for 1, 3, or 6 h.

### RNA expression analysis

Total RNA was extracted using TriZol Reagent (Invitrogen). First-strand cDNA synthesis was performed on 1.5 µg DNAse I (Sigma-Aldrich) treated RNA using poly (dT) primers with First Strand cDNA Synthesis kit (Thermo Fisher Scientific) as per manufacturer's protocol. Real time qPCR was performed using Master Mix Taqman Universal II (Thermo Fisher Scientific) on QuantStudio 3 (Applied BioSystems). HPRT1 was used for normalization. Primer sequences are given in Table 2.

### Protein expression analysis

Nuclear protein fraction was performed using the NE-PER Nuclear and Cytoplasmic extraction reagent (Thermo Fisher Scientific) by following manufacturer's protocol. RIPA-soluble and insoluble fractions were performed by lysing cells with RIPA buffer (Millipore) and 1X Halt Protease and Phosphatase inhibitor cocktail (Thermo Fisher Scientific). Protein cell lysates were centrifuged at 15,000$g$ at 4°C for 15 min. The supernatant was collected as RIPA-soluble protein. The RIPA-insoluble (urea) fraction was prepared as described (Hans et al, 2020). Briefly, the pellet was resuspended in 8 M urea buffer (10 mM Tris, pH = 8.0, 100 mM NaH$_2$PO$_4$, 8 M urea) containing 1X protease inhibitor, then vortexed every 10 min for 40 min before being centrifuged at 15,000$g$ for 15 min at 4°C. Proteins were quantified by the BCA Protein Assay (Pierce) or Bradford assay (Dutscher). 15 µg of protein homogenates were loaded on polyacrylamide gels and electroblotted onto nitrocellulose membranes (Invitrogen). After the total protein staining (LICORbio), membranes were blocked, incubated overnight at 4°C with one of the following primary antibodies: rabbit anti-TDP-43, rabbit anti-STMN2 or mouse anti-α-tubulin. The membranes were washed and incubated with anti-rabbit or anti-mouse infrared secondary antibodies (LICORbio). Detection was conducted using the Odyssey CLx imager (LICORbio).

## In vivo SOD1$^{G93A}$ mice experiment

### SOD1$^{G93A}$ mouse model

All experiments were carried out in accordance with the Italian Governing Law and followed current European Union regulations (Directive 2010/63/EU). Experimental procedures were approved by the National and Institutional Ethical Committees. Female transgenic SOD1$^{G93A}$ mice on C57BL/6J genetic background were purchased from the Jackson Laboratory (stock 004435, B6 Cg-Tg (SOD1*G93A)1Gur). Mice were maintained at a temperature of 22°C ± 2°C with a relative humidity of 55% ± 10% and 12 h light/dark cycle. Food (standard pellets) and water were supplied ad libitum.

### Treatment schedule and tissue collection

The mice were randomly divided into 3 groups of treatment and received Sephin1 4 mg/kg (n = 22), Sephin1 8 mg/kg (n = 22), or vehicle (saline solution; n = 22) by oral daily gavage. Treatment was performed starting from 8 wk of age (58 d old; before symptom onset) until 20 wk of age, when they were euthanized, plasma and spinal cord were collected for biochemical and histological analyses (n = 9), or until end-stage (n = 13), to evaluate the effect of the treatments on survival according to the guidelines (Ludolph et al, 2010). For biochemical and molecular analyses, five mice in each group were deeply anesthetized and tissues were rapidly removed, frozen in cold isopentane on dry ice and stored at –80°C. For the histopathology and immunohistochemistry analysis, four mice in each group were euthanized under deep anesthesia by intracardiac perfusion with fixative solution and tissues were rapidly removed, post-fixed for 2 h and then cryoprotected in sucrose solution overnight before being frozen in isopentane on dry ice.

### Disease onset and survival evaluation

The onset of symptoms was considered when the mice show the first impairment at the paw grip endurance test. The final stage of the disease was defined as the point at which the mice were no longer right themselves within 10 s after being placed on their sides. This time point was considered as the measure of survival duration.

### Paw grip endurance and Rotarod tests

The animals were placed on a horizontal grid, which was then inverted, allowing the mice to hang upside down. The time spent hanging, for a maximum of 90 s, before falling from the grid, was considered for the analysis. Mice that fell before 90 s repeated the trial, for a maximum of three attempts and a minimum 3 min of rest between each. The final score was calculated based on the maximum latency to fall.

The rotarod test was performed using the Rota-Rod apparatus (model 7650; Ugo Basile). The animals were placed on the rotating bar with a constant acceleration of four rotations per minute (rpm) starting from a speed of 7 rpm up to 28 rpm for a maximum of 300 s. The latency to fall was recorded and the test was repeated for a maximum of two times interspersed by 5 min rest between sessions when the mice did not reach the 300 s. The highest recorded latency was used for analysis.

### Motor neuron count

Choline O-acetyltransferase (ChAT) immunostaining was performed on 30 $\mu$m spinal cord sections (segments L2–L5, one every 10 sections) using a primary anti-ChAT antibody and a fluorescent secondary antibody, donkey anti-goat Alexa 647 (Life Technologies) following a standard immunofluorescence protocol. The sections were examined using an Olympus Fluoview confocal microscope (Olympus) and ChAT-immunopositive neurons with a cell body area >400 $\mu$m$^2$ were counted in each hemisection.

### Western blotting

Spinal cord was homogenized in ice-cold homogenization buffer (RIPA buffer and protease and phosphates inhibitor cocktail by Roche), centrifuged at 12,000$g$ for 30 min at 4°C and the supernatants were collected and stored at –80°C. For the extraction of Detergent-Insoluble Fraction (TIF), spinal cords were processed as previously described (Basso et al, 2009). Briefly, they were homogenized in ice-cold homogenization buffer, pH 7.6, containing 15 mM Tris–HCl, 1 mM DTT, 0.25 M sucrose, 1 mM MgCl$_2$, 2.5 mM EDTA, 1 mM EGTA, 0.25 M sodium orthovanadate, 2 mM sodium pyrophosphate, 5 $\mu$M MG132 proteasome inhibitor (Sigma-Aldrich), 1 tablet of Complete/10 ml of buffer, Mini Protease Inhibitor Mixture (Roche Applied Science). The samples were centrifuged at 10,000$g$ at 4°C for 15 min, obtaining a supernatant (S1) and a pellet. The pellet was suspended in ice-cold homogenization buffer with 2% of Triton X-100 and 150 mM KCl, sonicated three times for 10 s and shaken for 1 h at 4°C. Samples were then centrifuged twice at 10,000$g$ at 4°C for 10 min to obtain Triton X-100-resistant pellets (TIF) and a supernatant (S2). The soluble fraction is considered the pool of S1 and S2 fractions. Proteins were quantified by the Bradford assay. Equal amounts of total protein homogenates were loaded on polyacrylamide gels and electroblotted onto PVDF membranes (Millipore). After blocking, membranes were incubated overnight at 4°C with one of the following primary antibodies: rabbit anti-eIF2$\alpha$ rabbit, anti-phospho-eIF2$\alpha$, rabbit anti-hSOD1 or rabbit anti-TDP-43 (Table 1). The membranes were washed and incubated with horseradish peroxidase-conjugated anti-rabbit secondary antibodies (Santa Cruz) and developed by Luminata Forte Western Chemiluminescent horse radish peroxidase (HRP) Substrate (Millipore) on the Chemi-Doc XRS system (Bio-Rad). Densitometric analysis was performed using ImageLab (Bio-Rad) software.

### Plasmatic dosage of NFL

The NFL level in plasma was measured using the Simoa NF-light Advantage (SR-X) Kit (#103400) on the Quanterix SR-X platform according to the protocol issued by the manufacturer (Quanterix Corp).

### In vivo mutated TDP-43 transgenic zebrafish experiment

### Zebrafish model, treatment schedule, and tissue collection

All experiments were carried out in accordance with the French National Institutes of Health Guide for the Care and Use of Laboratory Animals and followed current European Union regulations (Directive 2010/63/EU). Experimental procedures were approved by the National and Institutional Ethical Committees.

### Zebrafish maintenance

Adult and larval zebrafish (*Danio rerio*) were maintained at the Imagine Institute (Paris, France) fish facilities and bred according to the National and European Guidelines for Animal Welfare. Experiments were performed on wild type and transgenic embryos from AB strains. Zebrafish were raised in embryo medium: 0.6 g/liter aquarium salt (Instant Ocean, Blacksburg, VA) in reverse osmosis water 0.01 mg/liter methylene blue. Embryos were staged in terms of hours post fertilization (hpf) based on morphological criteria (Westerfield, 2007) and manually dechorionated using fine forceps at 24 hpf. All the experiments were conducted on morphologically normal embryos.

### Mutant TDP-43 transgenic zebrafish line

The mutant TDP-43 transgenic line was previously characterized (Lissouba et al, 2018). Briefly, a construct for the human TARDBP cDNA bearing the G1176T change encoding for the G348C mutation was inserted in the Tol2Kit to generate the pDest-Tol2CG2-hsp70-TDP43-Myc vectors also containing the cardiac muscle promoter (cmlc2) expressing eGFP for detection of the transgene. This construct was injected into one-cell stage zebrafish embryos with 25 ng/$\mu$l of the Tol1 transposase mRNA with 0.05% Fast Green (Sigma-Aldrich). Approximately 300 embryos for this line were raised with <40% expressed eGFP in the heart. These embryos were selected and raised to adulthood as the F0 founders.

### Drug testing and heat shock procedure

Toxicity assessment was performed for Sephin1 with the 10 $\mu$M concentration not leading to any developmental delays or mortality observed in non-transgenic embryos incubated from 32 to 48 hpf. For the heat shock procedure, 30 hpf transgenic and non-transgenic embryos were dechorionated and screened for eGFP in the heart. These embryos were transferred to glass tubes with system water and then placed in a preheated water bath at 39°C for 10 min at 32 hpf. Transgenic and WT embryos were then placed in six well plates containing Sephin1 dissolved in DMSO at 10 $\mu$M as well as the DMSO control conditions and put back at 28.5°C. Zebrafish embryos received a second heat shock of 39°C for 1 h at 48 hpf. These experiments were replicated in at least three independent experiments.

### Motor behavior of zebrafish embryos

Locomotor phenotypes of 50 hpf zebrafish embryos were assessed using the *Touched-Evoked Escape Response* (TEER) test, as previously described (Ciura et al, 2013; Lattante et al, 2015; de Calbiac et al, 2018). Briefly, embryos were touched on the tail with a tip and the escape response was recorded using a Grasshopper 2 camera (Point Grey Research) at 30 frames per second. Distance and velocity parameters were quantified per each embryo using the video tracking plugin of FIJI 1.47 software (Schindelin et al, 2012). Spontaneous movements of 50 hpf zebrafish embryos were analyzed using an automated imaging and analysis system (Zebralab, ViewPoint). Single embryos were placed in individual well of a 96 well plate and recorded. Distance and velocity parameters were computed using the Live Tracking module in 20 min intervals.

### Motor neuron morphology and survival

Mutant TDP-43 transgenic zebrafish were crossed with *Tg(UAS:RFP)* zebrafish to generate double transgenic zebrafish line having a specific expression of RFP in motor neurons allowing the observation of cell bodies of single motor neurons and their axonal arborization within a somatic segment in fixed and lived animals. These embryos were used to image and quantify motor neurons features. For axonal projection measurements and cell body counting, 51 hpf embryos were fixed in 4% PFA and captured at the same defined location within the intersomitic segments with an Apotome.2 system and an Imager.M2 stand (Carl Zeiss), with a 20X objective (NA0.8). Motor neuron axonal length normalized to the spinal cord thickness was measured with FIJI 1.47 (Schindelin et al, 2012). Fluorescent cell bodies of motor neurons were counted within the same intersomitic segment region.

### Western blot TDP-43

Briefly, about 30 embryos for each condition were lysed in ice-cold RIPA buffer (150 mM NaCl, 50 mM Tris pH 7.5, 1% Triton X-100, 0.1% SDS, 1% Na deoxycholate, 0.1% protease inhibitor), maintained on ice and homogenized with a hand-held pestle. The lysates were centrifuged for 10 min at 7,500*g* at 4°C and the supernatants were collected. Protein concentration was established using Bio-Rad DC Protein Assay, 60 μg of proteins were loaded in 2X Laemmli buffer after boiling the samples for 5 min at 95°C on acrylamide gel and transferred on PVDF membrane. After blocking, membranes were incubated overnight at 4°C with one of the following primary antibodies: rabbit anti-TDP-43 or mouse anti-actin. The membranes were washed and incubated with anti-rabbit or anti-mouse infrared antibodies (LICORbio). Detection was conducted using the Odyssey CLx imager (LICORbio).

### Statistical analysis

All values are expressed as mean ± SEM (standard error of the mean). Statistical analysis was performed by D'Agostino and Pearson test for normality test, then by the Mann-Whitney test or unpaired *t* test for comparing two conditions or mixed effects analysis followed by Tukey's multiple comparisons test, or by Kruskal-Wallis test followed by Dunn's multiple comparison test for more than two conditions comparison (Fig S6A). For longitudinal studies, statistical analysis was performed by two-way ANOVA followed by Tukey multiple comparison test for comparing more than two timepoints and two conditions or by Friedman test followed by Dunn's multiple comparison test to compare more than two timepoints for one treatment (Fig S6B). $P < 0.05$ was considered significant. Statistical analysis was performed on GraphPad Prism software version 10.3.1.

## Data Availability

Data supporting these findings and Sephin1 are available from the corresponding authors upon request.

## Supplementary Information

## Acknowledgements

We thank Dr Catherine Botto and the technical department of Neuro-Sys SAS for their technical support in conducting all in vitro experiments.

### Author Contributions

E Abgueguen: conceptualization, formal analysis, supervision, investigation, visualization, methodology, project administration, and writing—original draft, review, and editing.

M Tortarolo: investigation, validation, and writing—review and editing.

L Rouviere: investigation and validation.

S Marcuzzo: methodology, investigation, and writing—review and editing.

L Camporeale: investigation and validation.

A Henriques: investigation, methodology, and validation.

L Pasetto: investigation and validation.

GR Culley: investigation and writing—review and editing.

V Bonetto: investigation and validation.

A Marian: investigation and methodology

BL Lejeune: conceptualization, project administration, and writing—review and editing.

A Visbecq: conceptualization and project administration.

G Lauria: conceptualization and writing—review and editing.

E Kabashi: conceptualization, investigation, methodology, validation, and writing—review and editing.

N Callizot: conceptualization, investigation, methodology, and writing—review and editing.

C Bendotti: conceptualization, investigation, methodology, and writing—review and editing.

PY Miniou: conceptualization, supervision, funding acquisition, project administration, and writing—review and editing.

### Conflict of Interest

E Abgueguen, BL Lejeune, A Visbecq, and PY Miniou are employed by InFlectis BioScience. PY Miniou is co-founder of InFlectis BioScience. N Callizot is a member of the scientific advisory board and consultant for InFlectis BioScience.

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
