## [Reviewer comments · Life Science Alliance]

Life Science Alliance

Sephin1 reduces TDP-43 cytoplasmic mislocalization and improves motor neuron survival in ALS models

Emmanuelle Abgueguen, Massimo Tortarolo, Laura Rouvière, Stefania Marcuzzo, Laura Camporeale, Alexandre Henriques, Laura Pasetto, Georgia Culley, Valentina Bonetto, Beatrice Lejeune, Anne Visbecq, Giuseppe Lauria, Edor Kabashi, Noëlle Callizot, Caterina Bendotti, and Pierre Miniou

DOI: <https://doi.org/10.26508/lsa.202403195>

Corresponding author(s): *Emmanuelle Abgueguen, InFlectis BioScience and Pierre Miniou, InFlectis BioScience*

Review Timeline:

Submission Date:	2024-12-26
Editorial Decision:	2025-01-31
Revision Received:	2025-05-19
Editorial Decision:	2025-06-16
Revision Received:	2025-06-20
Accepted:	2025-06-20

Scientific Editor: Tim Fessenden

Transaction Report:

January 31, 2025

Re: Life Science Alliance manuscript #LSA-2024-03195-T

Dr. Emmanuelle Abgueguen
InFlectis BioScience
21 rue La Noué Bras de fer, 44200 Nantes, France
Nantes, Autre 44200
France

Dear Dr. Abgueguen,

Thank you for submitting your manuscript entitled "Sephin1 improves motor neuron survival in ALS models by reducing TDP-43 cytoplasmic mislocalization". The manuscript has been evaluated by expert reviewers, whose reports are appended below. Unfortunately, after an assessment of the reviewer feedback, our editorial decision is against publication in Life Science Alliance.

Although your manuscript is intriguing, I feel that the points raised by the reviewers are more substantial than can be addressed in a typical revision period. If you wish to expedite publication of the current data, it may be best to pursue publication at another journal.

Given the interest in the topic, I would be open to re-submission to Life Science Alliance of a significantly revised and extended manuscript that fully addresses the reviewers' concerns and is subject to further peer review. If you would like to resubmit this work to Life Science Alliance, you may submit an appeal directly through our manuscript submission system. Please note that priority and novelty would be reassessed at re-submission.

Regardless of how you choose to proceed, we hope that the comments below will prove constructive as your work progresses.

Thank you for thinking of Life Science Alliance as an appropriate place to publish your work.

Sincerely,

Reviewer #1 (Comments to the Authors (Required)):

Summary:

Abgueguen and collaborators tested the hypothesis that Sephin1, a selective inhibitor of the phosphatase regulatory subunit PPP1R15A and modulator of the UPR pathway, could reduce TDP-43 pathology and therefore could be beneficial for several ALS models, both in vitro and in vivo. The authors showed that Sephin1 improves the survival of SOD1G93A primary motor neurons exposed to glutamate excitotoxicity and protects the neurite network. The authors examined the mechanisms implicated in the beneficial effects in vitro and found that Sephin1 significantly reduces mitochondrial ROS and extranuclear TDP-43 localization in SOD1G93A primary motor neurons following glutamate excitotoxicity. Next, the authors tested the drug in vivo, on ALS mice overexpressing the SOD1G93A mutant gene. Although the authors observed a significant effect on motor neuron survival, they were unable to reproduce the previously published beneficial effects on motor function and bodyweight. Moreover, the authors did not observe an increased overall survival. When treating a TDP-43 zebrafish model of ALS, this time the authors reported significant beneficial effects of Sephin1 on survival and motor functions.

The significance of this research lies in its demonstration that Sephin1 can mitigate TDP-43 pathology, which is present in about 97% of ALS patients. This offers a novel therapeutic approach for ALS and potentially other TDP-43 proteinopathies. While the manuscript is clear and well written, the data are not always strongly supporting the conclusions, and some additional experiments would benefit the study and the field of ALS research.

Comments:

The reason why the authors decided to work with SOD1 models of ALS in their study is not clear. The drug has already been extensively tested with these models and the presence of TDP-43 aggregates in human postmortem materials remains highly debated. It is clear that in the spectrum of TDP-43 proteinopathies, SOD1-ALS patients lie apart. Could the authors explain the relevance of such models?

1. Sephin1 protects motor neurons against glutamate excitotoxicity

It is important to add representative images of the neurite network. The reader should be able to assess how the Sephin1 treatment affects the neurite integrity. And in the Y axis of the neurite network in should rather be noted total neurite length, which is what is measured. The motor neuron viability is based on the number of MAP-2 positive cells. This can be biased by different seeding efficacy. A caspase assay for apoptosis would have been more suitable, especially because the cells are treated with glutamate, which is known to activate apoptosis.

The final sentence of the conclusion should be "this increase of survival is accompanied by a protection of the neurite network, only in SOD1G93A primary motor neurons."

2. Sephin1 reduces mitochondrial ROS without modulating calcium influx and eIF2 α phosphorylation level following glutamate intoxication in primary motor neurons

Representative images of the Fluo 4AM are missing. Similarly, I am missing the western blot images accompanying fig2B and representative images of the mitochondrial ROS staining for Fig2C. Please note that the MitoSOX dye used in the study is only oxidized by superoxide and not by other ROS. This is important to note in the manuscript. Overall, quantification should be better explained in the methods.

3. Sephin1 reduces extranuclear TDP-43 in glutamate intoxicated primary motor neurons

This is the most important part of the manuscript and I am not convinced by the methods. How did the author quantify what is extranuclear TDP-43? Is it diffuse signal in the cytoplasm or aggregates? Both? If it is aggregates, a better characterization is needed. For instance, co-labelling with P62. The images presented are not convincing. There are with pore resolution and should be taken at higher magnification. Also how pure is the cell culture? Why not using a ChAT marker for motor neurons? With a better purity the authors could have performed nucleo-cytoplasmic fractionnement and western blot to determine the percentage of TDP-43 in the cytoplasm.

How the authors explain that there is a ROS reduction at 500nM and no reduction of extranuclear TDP-43 at the concentration of Sephin1?

4. Sephin1 protects motor neurons and reduces TDP-43 aggregates in the spinal cord of SOD1G93A mouse model

The authors could add the gel images for fig4F and 4G. An immunostaining of TDP-43 and pTDP-43 in the spinal cord would have been important to confirm the trend observed in Fig4H-I.

5. Sephin1 improves survival and motor function in mutated TDP-43 transgenic zebrafish

Images with better resolution are needed.

Additional issues:

- Do you have the ethical approval numbers for all experiments?
- There is a type in the first line of the description of the protein analysis in the methods.
- In the material and methods, it is not clear to what refers the "as previously described" in the section of the western blot TDP-43. Please be more specific and add some details.
- Please, harmonized all graphs with data points within the bars.
- For clarity, it would be nice to precise directly of the graphs if the results come from WT or SOD1G93A cultures.
- In Fig4E it is written IFB-088 instead of Sephin. Similarly, it is written page 16.
- Neither the results part or the figure legends help to understand how the experiment has been performed. This is a recurrent problem of the manuscript. The results part would benefit from more technical details on how the experiments have been performed.
- It would be interesting to discuss how the treatment influence ROS production.
- Why there is no increase of eIF2 α in primary motor neurons and a dose dependent increase in the spinal cord?

Reviewer #2 (Comments to the Authors (Required)):

The authors performed in vitro studies using primary motor neurons from WT and SOD1G93A rats with glutamate excitotoxicity and in vivo studies using SOD1G93A mice and zebrafish embryos expressing mutant TDP-43G348C protein to investigate the role of Sephin1 in ALS. In this report, the authors presented their data. They show that Sephin1, at concentrations below 500 nM, improves the survival of glutamate-intoxicated primary motor neurons from WT and SOD1G93A rats. This is accompanied by protection of the neurite network, particularly in the latter. Sephin1 reduces mitochondrial ROS but has no impact on calcium influx or eIF2 α phosphorylation in primary motor neurons. Also, in both WT and SOD1G93A rat motor neurons, Sephin1 reduces extranuclear TDP-43 localization after glutamate intoxication. The authors also show that Sephin1 improves motor neuron survival and reduces TDP-43 aggregation but has no impact on motor function and overall survival in the SOD1G93A mouse model. Lastly, Sephin1 treatment improves survival, locomotor function, motor neuron survival, and axonal length of the zebrafish embryos expressing mutant TDP-43G348C protein. Overall, the data is of interest to the ALS research community. Key weaknesses are a lack of transparent presentation and overconclusion. The points that the authors need to address are outlined below:

1. In Fig. 2, the authors did not show primary data for how Sephin1 modulates calcium influx in primary SOD1G93A motor neurons. The authors also did not show primary data for how Sephin1 modulates the level of eIF2 α phosphorylation and mitochondrial ROS in primary WT motor neurons. For consistency, the authors should provide data for both primary WT and SOD1G93A motor neurons.
2. In Fig.3A, the authors showed image for with 100nM Sephin1, whereas significance was only seen at 500nM. The authors should show the 500 nM data for consistency.

3. The authors should provide a representative image of TDP-43 localization in SOD1G93A rat motor neurons in Fig. 3.
4. The authors should state the rationale for using only female SOD1G93A mice and discuss the tenuous nature of their conclusions from using few animals and a single sex.
5. In result section 5, the authors state, "IFB-088 does not modulate the intensity of this band either (Sup Fig 3A)". IFB-088 should be changed to Sephin1 for consistency.
6. The authors conclude by stating, "Overall, this study demonstrates that Sephin1 improves motor neuron survival in multiple models of ALS by reducing TDP-43 cytoplasmic mislocalization and TDP-43 aggregates, suggesting that Sephin1 could be a therapeutic strategy for ALS patients.". There is no direct evidence in this study to suggest reduced TDP-43 cytoplasmic mislocalization and TDP-43 aggregation are responsible for improved motor neuron survival. To support this statement, the authors need to provide evidence.
7. Considering the reduction of TDP-43 cytoplasmic mislocalization was only shown in the in vitro studies but not in the SOD1G93A mice and mutant TDP-43G348C zebrafish embryos, the title of the article overstates the data and should be revised.

Sephin manuscript, reviewers' comments and reply:

Reviewer #1 (Comments to the Authors (Required)):

Summary:

Abgueguen and collaborators tested the hypothesis that Sephin1, a selective inhibitor of the phosphatase regulatory subunit PPP1R15A and modulator of the UPR pathway, could reduce TDP-43 pathology and therefore could be beneficial for several ALS models, both in vitro and in vivo. The authors showed that Sephin1 improves the survival of SOD1G93A primary motor neurons exposed to glutamate excitotoxicity and protects the neurite network. The authors examined the mechanisms implicated in the beneficial effects in vitro and found that Sephin1 significantly reduces mitochondrial ROS and extranuclear TDP-43 localization in SOD1G93A primary motor neurons following glutamate excitotoxicity. Next, the authors tested the drug in vivo, on ALS mice overexpressing the SOD1G93A mutant gene. Although the authors observed a significant effect on motor neuron survival, they were unable to reproduce the previously published beneficial effects on motor function and bodyweight. Moreover, the authors did not observe an increased overall survival. When treating a TDP-43 zebrafish model of ALS, this time the authors reported significant beneficial effects of Sephin1 on survival and motor functions.

The significance of this research lies in its demonstration that Sephin1 can mitigate TDP-43 pathology, which is present in about 97% of ALS patients. This offers a novel therapeutic approach for ALS and potentially other TDP-43 proteinopathies. While the manuscript is clear and well written, the data are not always strongly supporting the conclusions, and some additional experiments would benefit the study and the field of ALS research.

Comments:

The reason why the authors decided to work with SOD1 models of ALS in their study is not clear. The drug has already been extensively tested with these models and the presence of TDP-43 aggregates in human postmortem materials remains highly debated. It is clear that in the spectrum of TDP-43 proteinopathies, SOD1-ALS patients lie apart. Could the authors explain the relevance of such models?

Thank you for your insightful comment. We acknowledge the ongoing debate regarding the presence of TDP-43 pathology in SOD1-ALS patients and mouse models. However, as this pathology is not definitively excluded, we believe it remains relevant to investigate potential mechanistic overlaps. The SOD1^{G93A} rodent model continues to be the most well-characterized and widely used model for ALS, faithfully recapitulating key neuropathological features of the disease. Additionally, while the drug has been previously tested in this mouse model, the reported results have been quite controversial. Therefore, we deemed it crucial to conduct further experiments to clarify its effects. This justification has been explicitly stated in the manuscript's introduction.

1. Sephin1 protects motor neurons against glutamate excitotoxicity

It is important to add representative images of the neurite network. The reader should be able to assess how the Sephin1 treatment affects the neurite integrity. And in the Y axis of the neurite network in should rather be noted total neurite length, which is what is measured. The motor neuron viability is based on the number of MAP-2 positive cells. This can be biased by different seeding efficacy. A caspase assay for apoptosis would have been more suitable, especially because the cells are treated with glutamate, which is known to activate apoptosis.

The final sentence of the conclusion should be "this increase of survival is accompanied by a protection of the neurite network, only in SOD1^{G93A} primary motor neurons."

Thank you for your comments. Representative pictures of WT and SOD1^{G93A} primary motor neurons labelled with MAP-2 antibody have been added to the manuscript allowing the readers to assess the viability and the neurite length of the MAP-2 labelled cells. The graphs representing the neurite network in percentage have been replaced by the graphs representing the neurite length in μm . Regarding the measure of cell viability, our primary motor neurons cultures contain motor neurons and glial cells as you can see on the representative picture in Figure 1. To measure specifically the motor neurons viability following glutamate intoxication, we decided to count the number of MAP-2 labelled cells per well instead of using a caspase assay. To avoid a bias due to a different seeding efficacy, the cells from the same isolation were seeded in 96 well plate, then 6 wells were used per tested condition. The result of cleaved caspase 3 expression level obtained by western blot on the primary motor neurons enriched cultures has been added to the manuscript. The final sentence of the conclusion has been modified according to your suggestion.

2. Sephin1 reduces mitochondrial ROS without modulating calcium influx and eIF2 α phosphorylation level following glutamate intoxication in primary motor neurons

Representative images of the Fluo 4AM are missing. Similarly, I am missing the western blot images accompanying fig2B and representative images of the mitochondrial ROS staining for Fig2C. Please note that the MitoSOX dye used in the study is only oxidized by superoxide and not by other ROS. This is important to note in the manuscript. Overall, quantification should be better explained in the methods.

Thank you for your comments. The level of intracellular Ca²⁺ was evaluated by measuring the fluorescence (EX/EM 494/508) with a plate reader, 5 minutes after glutamate application. The signal was measured on all the cells from the culture. No picture of Fluo 4AM was taken. Representative western blot images of p-eIF2A/total eIF2A and representative images of mitochondrial ROS staining were added to the manuscript. The procedures used to quantify cell viability, neurite length, mitochondrial ROS, calcium flux and mitochondrial ROS are better explained in this version of manuscript.

3. Sephin1 reduces extranuclear TDP-43 in glutamate intoxicated primary motor neurons

This is the most important part of the manuscript and I am not convinced by the methods. How did the author quantify what is extranuclear TDP-43? Is it diffuse signal in the cytoplasm or aggregates? Both? If it is aggregates, a better characterization is needed. For instance, co-labelling with P62. The images presented are not convincing. There are with pore resolution and should be taken at higher magnification. Also how pure is the cell culture? Why not using a ChAT marker for motor neurons? With a better purity the authors could have performed nucleo-cytoplasmic fractionnement and western blot to determine the percentage of TDP-43 in the cytoplasm.

How the authors explain that there is a ROS reduction at 500nM and no reduction of extranuclear TDP-43 at the concentration of Sephin1?

Thank you for your comments. The extranuclear localization of TDP-43 is estimated by immunolocalization 24 hours after glutamate intoxication. The area overlapping TDP-43 and MAP-2 immunostaining and excluding the area overlapping Hoechst (nucleus) and TDP-43 staining is considered as TDP-43 extranuclear localisation. The signal of extranuclear TDP-43 could then be diffused or more pronounced like in aggregates. The extranuclear TDP-43 measurement does not consider whether TDP-43 is present in aggregates or in stress granules in the cytoplasm. We only

wanted to determine if Sephin1 was able to reduce the presence of TDP-43 in the cytoplasm. This description has been explicitly stated in the manuscript's result.

In the primary motor neurons experiment, we used a motor neuron enriched culture. This culture contains glia cells and motor neurons preventing us to perform nuclear and cytoplasmic fractions and then to determine by western blot TDP-43 level in the cytoplasm.

The ROS reduction observed at 500nM was performed in primary SOD1^{G93A} motor neurons (Figure 2F-G). In these cells, we observed a trend to a reduction of TDP43 at this concentration (Figure 3C). The addition of the SOD1 or WT motor neurons in the Y axis of the graph should avoid any confusion.

4. Sephin1 protects motor neurons and reduces TDP-43 aggregates in the spinal cord of SOD1G93A mouse model

The authors could add the gel images for fig4F and 4G. An immunostaining of TDP-43 and pTDP-43 in the spinal cord would have been important to confirm the trend observed in Fig4H-I.

Thank you for your valuable suggestion. We have added the representative blot images for Fig. 4F and 4G as requested. Regarding the immunostaining of TDP-43 and pTDP-43 in the spinal cord, while it can provide qualitative insights, it is inherently less reliable for quantitative analysis. Based on the existing literature, we believe that the western blot data offer a more robust and reproducible assessment of TDP-43 levels.

5. Sephin1 improves survival and motor function in mutated TDP-43 transgenic zebrafish

Images with better resolution are needed.

Additional issues:

- Do you have the ethical approval numbers for all experiments?

We can confirm that we received ethical approval by the institutional and ministerial ethics committees for the experiments done on primary motor neurons performed by NeuroSys, the SOD1^{G93A} mice performed in the Caterina Bendotti's lab and the zebrafish performed at the Imagine Institute.

-

There is a type in the first line of the description of the protein analysis in the methods.

- In the material and methods, it is not clear to what refers the "as previously described" in the section of the western blot TDP-43. Please be more specific and add some details.

- Please, harmonized all graphs with data points within the bars.

- For clarity, it would be nice to precise directly of the graphs if the results come from WT or SOD1G93A cultures.

- In Fig4E it is written IFB-088 instead of Sephin. Similarly, it is written page 16.

Thank you for your comments. The modifications were made in the manuscript.

- Neither the results part or the figure legends help to understand how the experiment has been performed. This is a recurrent problem of the manuscript. The results part would benefit from more technical details on how the experiments have been performed.

Thank you for your suggestions. A brief description of the experiments was added in the result section. A better explanation of the procedures used to quantified cell viability, neurite length, calcium flux and mitochondrial ROS have been added to the materiel and method section.

- It would be interesting to discuss how the treatment influence ROS production.

Currently, we do not have identified, yet, the mechanism of action leading to reduce ROS production. In the clinical trial, we have observed a reduction of oxidative stress markers in ALS patients treated for 6 months with Sephin1. So, we believe that Sephin1 reduces oxidative stress but we don't know how. The result of the clinical trial will be presented at the ENCALS meeting in June 2025. A manuscript is in preparation to present the clinical trial results. We hope it will be published in few months.

- Why there is no increase of eIF2 α in primary motor neurons and a dose dependent increase in the spinal cord?

Thank you for your comment. The absence of a detectable increase in eIF2 α levels in primary motor neurons, despite a dose-dependent increase observed in the spinal cord, could be attributed to the involvement of multiple cell types in the spinal cord tissue.

It is known that eIF2 α phosphorylation is not restricted to neurons but is also activated in microglia in response to stress conditions (Ruggeri 2023). Recently it has been reported that the stress-induced stress granule assembly through the phosphorylation of eIF2 α in microglia, precludes the activation of NLRP3 inflammasome, becoming neuroprotective (Wu et al. 2023). Thus, the increase of eIF2 α phosphorylation observed in spinal cord may be due to the activated microglia typical of mouse and human ALS spinal cord. This has been included in the discussion.

Reviewer #2 (Comments to the Authors (Required)):

The authors performed in vitro studies using primary motor neurons from WT and SOD1G93A rats with glutamate excitotoxicity and in vivo studies using SOD1G93A mice and zebrafish embryos expressing mutant TDP-43G348C protein to investigate the role of Sephin1 in ALS. In this report, the authors presented their data. They show that Sephin1, at concentrations below 500 nM, improves the survival of glutamate-intoxicated primary motor neurons from WT and SOD1G93A rats. This is accompanied by protection of the neurite network, particularly in the latter. Sephin1 reduces mitochondrial ROS but has no impact on calcium influx or eIF2 α phosphorylation in primary motor neurons. Also, in both WT and SOD1G93A rat motor neurons, Sephin1 reduces extranuclear TDP-43 localization after glutamate intoxication. The authors also show that Sephin1 improves motor neuron survival and reduces TDP-43 aggregation but has no impact on motor function and overall survival in the SOD1G93A mouse model. Lastly, Sephin1 treatment improves survival, locomotor function, motor neuron survival, and axonal length of the zebrafish embryos expressing mutant TDP-43G348C protein. Overall, the data is of interest to the ALS research community. Key weaknesses are a lack of transparent presentation and overconclusion. The points that the authors need to address are outlined below:

1. In Fig. 2, the authors did not show primary data for how Sephin1 modulates calcium influx in primary SOD1G93A motor neurons. The authors also did not show primary data for how Sephin1 modulates the level of eIF2 α phosphorylation and mitochondrial ROS in primary WT motor neurons. For consistency, the authors should provide data for both primary WT and SOD1G93A motor neurons.

Thank you for your valuable suggestion. As we did not observe an increase of calcium flux by Sephin1 treatment in primary WT motor neurons, we did not expect to observe one in primary SOD1^{G93A} motor neurons. Regarding the increase of mitochondrial ROS, we observed an increase of

mitochondrial ROS in primary motor neurons from SOD1^{G93A} rat and in primary cortical neurons from WT rat. We decide to not present the data on mitochondrial ROS obtained on primary cortical neurons from WT rat because these data were observed when the cortical neurons were stimulated with NMDA and not glutamate.

The data on the eIF2a phosphorylation level in primary WT and SOD1^{G93A} motor neurons were added to the manuscript in the result section.

2. In Fig.3A, the authors showed image for with 100nM Sephin1, whereas significance was only seen at 500nM. The authors should show the 500 nM data for consistency.

Thank you for your comments. The images presented in Figure 3A are taken from primary SOD1^{G93A} motor neurons culture. In these cells, 100nM of Sephin1 significantly reduces extranuclear TDP-43 localisation (see Figure 3C). In WT motor neurons, 500µM Sephin 1 significantly extranuclear TDP-43 localisation (see Figure 3B). We did not add a representative image of TDP-43 in WT motor neurons in the ~~manuscript~~ manuscript.

3. The authors should provide a representative image of TDP-43 localization in SOD1^{G93A} rat motor neurons in Fig. 3.

Thank you for your valuable suggestion. We have added on the representative image “arrow” and “cross” to help the reader to observe the localisation of TDP-43 in the image.

4. The authors should state the rationale for using only female SOD1^{G93A} mice and discuss the tenuous nature of their conclusions from using few animals and a single sex.

Thank you for your comment. We originally used SOD1^{G93A} male mice exclusively for colony propagation, as females carrying the transgene are usually sterile or unable to carry a pregnancy. As a result, most of our accumulated data comes from female mice, and in order to maintain consistency in comparison with previous results we decided to do the study only in females. However, we fully acknowledge the importance of investigating both sexes, especially given the growing evidence suggesting sex-specific differences in disease mechanisms and therapeutic responses. We will ensure that this limitation is explicitly stated in the manuscript and discussed as a factor that should be addressed in future studies. In any case, we used a number of animals per group as suggested in the published guidelines (Ludolph et al. 2010), this has been explicated in the results.

5. In result section 5, the authors state, "IFB-088 does not modulate the intensity of this band either (Sup Fig 3A)". IFB-088 should be changed to Sephin1 for consistency.

Thank you for your comments. The change has been made.

6. The authors conclude by stating, "Overall, this study demonstrates that Sephin1 improves motor neuron survival in multiple models of ALS by reducing TDP-43 cytoplasmic mislocalization and TDP-43 aggregates, suggesting that Sephin1 could be a therapeutic strategy for ALS patients.". There is no direct evidence in this study to suggest reduced TDP-43 cytoplasmic mislocalization and TDP-43 aggregation are responsible for improved motor neuron survival. To support this statement, the authors need to provide evidence.

Thank you for your comment. Several publications have shown that the localisation of TDP-43 could modulate cell viability. In fact, elevated cytotoxicity is observed in cells and animal models expressing cytoplasmic TDP-43, whereas preventing TDP-43 nuclear export partially reduces TDP-43 cytotoxicity (Walker et al., 2015, *Acta Neuropathol*; Kabashi et al., 2010, *Hum Mol Genet*; Diaper et al., 2013, *Hum Mol Genet*; Cascella et al., 2016, *J Biol. Chem.*). Furthermore, loss of nuclear TDP-43 function leads to abnormal RNA splicing and incorporation of erroneous cryptic exons which provokes the loss of crucial neuronal proteins such as Stathmin 2 or UNC13A (Mehta et al., 2023 *Mol Neurodegenera*; Koike et al., 2024, *JMA J*).

In this manuscript, we show that Sephin1 reduces TDP-43 cytoplasmic localisation in primary motor neurons, reduces TDP-43 in Triton insoluble fraction in spinal cord of SOD1G93A mice and reduces TDP-43 toxicity in TDP-43 mutated zebrafish. In all ALS models, we observed an increase of motor neurons survival. To this version of the manuscript, we have added new data showing that Sephin1 could reduce cytoplasmic TDP-43 and abnormal splicing due to TDP-43 nuclear loss of function. We hope that the data can convince the reader that Sephin 1 can improve motor neuron survival by reducing cytoplasmic TDP-43.

For your information, in the clinical trial, we have observed a reduction of TDP-43 protein level in EVs from ALS patients treated for 6 months with Sephin1. EVs are small vesicles in a lipid bilayer containing protein, RNA and DNA and secreted from almost all cells and are detectable in plasma or CSF. EVs are able to cross the blood-brain barrier and are a means of intercellular signaling to and from the central nervous system. Several studies have shown that the level of TDP-43 in EVs from plasma or CSF are increased in ALS patients compared to healthy control which we confirmed in our study. The presence of TDP-43 in EVs likely indicates TDP-43 pathological relocation from the nucleus to the cytoplasm as nuclear export is a prerequisite for TDP-43 incorporation into EVs. So, we believe that Sephin1 reduces TDP-43 in cytoplasm. The result of the clinical trial will be presented at the ENCALS meeting in June 2025. A manuscript is in preparation to present the clinical trial results. We hope it will be published in few months.

7. Considering the reduction of TDP-43 cytoplasmic mislocalization was only shown in the in vitro studies but not in the SOD1G93A mice and mutant TDP-43G348C zebrafish embryos, the title of the article overstates the data and should be revised.

Thank you for your comment. We have modified the title of the article. The title "Sephin1 improves motor neuron survival in ALS by reducing TDP-43 cytoplasmic mislocalization" was replaced by "Sephin1 reduces TDP-43 cytoplasmic mislocalization and improves motor neurons survival in ALS models."

June 16, 2025

RE: Life Science Alliance Manuscript #LSA-2024-03195-TR-A

Dr. Emmanuelle Abgueguen
InFlectis BioScience
21 rue La Noué Bras de fer, 44200 Nantes, France
Nantes, Autre 44200
France

Dear Dr. Abgueguen,

Thank you for submitting your revised manuscript entitled "Sephin1 reduces TDP-43 cytoplasmic mislocalization and improves motor neuron survival in ALS models". As you will see, reviewers are overall satisfied, with minor corrections sought by Reviewer 2. We would be happy to publish your paper in Life Science Alliance pending these corrections and final revisions necessary to meet our formatting guidelines.

- Please be sure that the authorship listing and order is correct.
- Please upload your main manuscript text as an editable doc file.
- Please upload your main and supplementary figures as single files.
- Please add ORCID ID for secondary corresponding author--they should have received instructions on how to do so.
- Please add the X and Bluesky handles of your host institute/organization as well as your own or/and one of the authors in our system.
- On the title page of the manuscript, please provide each author's full name, including middle names as initials, formatted as follows: first name, middle initial, Last name.
- Please be sure that the authorship listing and order are correct and match between the system and the manuscript file.
- Please consult our manuscript preparation guidelines <https://www.life-science-alliance.org/manuscript-prep> and make sure your manuscript sections are in the correct order.
- The contributions selected for Stefania Marcuzzo, Georgia Culley, Beatrice Lejeune, Anne Visbecq and Guiseppe Lauria do not qualify them for authorship. Please either update the contributions in our system and the Author Contributions section of the manuscript, or let us know if the authors need to be removed (and added eventually to the acknowledgment section).
- Please upload your Tables in editable .doc or Excel format.
- Please add your main, supplementary figure, and table legends to the main manuscript text after the references section.
- Please use the [10 author names, et al.] format in your references (i.e., limit the author names to the first 10).
- We encourage you to revise the figure legends for Figure 8 such that the figure panels are introduced in alphabetical order.
- Please add a Data Availability section to the manuscript text.
- Please add callouts for Figure S5A-B to your main manuscript text.
- Please add molecular weight markers for protein blots in Fig 6 and Fig S2.
- Please ensure labels on the images in Fig 1 and Fig 2 are legible.
- Please add scale bars to images in Fig 5.

A. FINAL FILES:

B. MANUSCRIPT ORGANIZATION AND FORMATTING:

Sincerely,

Reviewer #1 (Comments to the Authors (Required)):

Abgueguen and collaborators submitted an improved revised manuscript. The authors addressed all my points accurately, incorporating modifications into the text, figures, and additional results. This results in a manuscript that is easier to understand and stronger. The experiment identifying the effect of Sepin1 on the splicing defects strengthens the story. Although the limitation of this experiment could have been described in the discussion. Thanks to the authors' efforts, I can validate the revisions and endorse the publication.

Reviewer #2 (Comments to the Authors (Required)):

The authors have improved the manuscript by providing more data to support their claims, have revised the manuscript title as suggested, and have provided convincing rebuttal where necessary. Although some typographical errors are noticed in the manuscript, I advise the manuscript to be accepted after the errors are fixed.

Below are some of the errors.

- In the subsection 4 of the result section, 4- Sephin 1 reduces abnormal splicing during stress recovery- the authors stated, "At protein level, STMN2 protein level is higher in sephin1 treated cells at 3 hours of recovery (Sup Fig. 2L)." There is no Sup Fig. 2L. Do the authors mean Sup Fig. 2K?

- In figure legend 2, "primary ray motor neurons" should be corrected to primary rat motor neurons.
- The authors should carefully go through the manuscript to correct other typographical errors.

June 20, 2025

RE: Life Science Alliance Manuscript #LSA-2024-03195-TRR

Dr. Emmanuelle Abgueguen
InFlectis BioScience
21 rue La Noué Bras de fer, 44200 Nantes, France
Nantes, Autre 44200
France

Dear Dr. Abgueguen,

Thank you for submitting your Research Article entitled "Sephin1 reduces TDP-43 cytoplasmic mislocalization and improves motor neuron survival in ALS models". It is a pleasure to let you know that your manuscript is now accepted for publication in Life Science Alliance. Congratulations on this interesting work.

DISTRIBUTION OF MATERIALS:

Again, congratulations on a very nice paper. I hope you found the review process to be constructive and are pleased with how the manuscript was handled editorially. We look forward to future exciting submissions from your lab.

Sincerely,
